# Spatial influence of agricultural residue burning and aerosols on land surface
temperature
Akanksha Pandey[1], Richa Singh[1], Kumari Aditi[1,2], Neha Chhillar[1], Tirthankar Banerjee[1,2*]
[1] Institute of Environment and Sustainable Development, Banaras Hindu University, Varanasi, India.
[2]DST-Mahamana Centre of Excellence in Climate Change Research, Banaras Hindu University, Varanasi, India.
*Correspondence to: T. Banerjee (tb.iesd@bhu.ac.in; tirthankaronline@gmail.com)
**Abstract**
The biophysical effects of agricultural residue burning, driven by the excessive release of
energy and carbonaceous aerosols, remain poorly quantified at the global scale. Residue-
based fires have the potential to modify regional climate by altering land surface temperature
(LST), highlighting the need for investigation at regional scale. Here, an observation-driven
assessment of spatial variations in LST due to concurrent release of energy and aerosols has
been made over northwestern India using multiple satellite and reanalysis-based datasets.
Year-specific fire pixel density was used to delineate an intensive fire zone characterized by
medium-to-large residue-based fire. Geospatial analysis revealed positive association among
FRP (fire radiative power), LST and AOD (aerosol optical depth). Over intensive fire zone, a
space-for-time approach revealed significant increase in both ΔLST (0.57°C; 95% CI:0.33-
0.81°C) and ΔAOD (0.13; 95% CI:0.08–0.17) due to fire. Random Forest non-linear model was
employed to regress potential influence of FRP and AOD on LST having several other variables
as confounding factors. FRP consistently emerged as the dominant predictor of LST, followed
by planetary boundary layer height and aerosols. An increase in relative feature importance
of FRP was noted during days having high fire intensity and positive association with LST.
Geographically weighted regression further explained spatial heterogeneity in LST
modulation by FRP. Overall, this analysis provides the first empirical evidence that residue-
based fire contributes to changes in land surface temperature. It further highlights that the
magnitude of this perturbation is governed by interannual variations in fire intensity and
influenced strongly by prevailing meteorological conditions.
**Keywords:** Aerosols, Biomass burning, Fire, GWR, Random Forest.

## Introduction

Burning agricultural residues is a widespread practice for the rapid removal of post-harvest biomass from croplands in many regions of the world (Streets et al., 2003; Singh et al., 2018; Shyamsundar et al., 2019). While biomass burning is often associated with deforestation (Chuvieco et al., 2021), forest fires (van der Velde et al., 2021; Aditi et al., 2025), and shifting cultivation (Prasad et al., 2000), residue burning on agricultural land is primarily conducted to clear fields, fertilize soil, eradicate weeds and pests, and prepare land for the next crop cycle (Graham et al., 2002; Korontzi et al., 2006; Lan et al., 2022). This practice is observed across large agricultural regions globally, including China (Streets et al., 2003; Zhang et al., 2020), South America (Graham et al., 2002), Southeast Asia (Lasko and Vadrevu, 2018; Yin, 2020), and northwestern India (Singh et al., 2018, 2021; Sarkar et al., 2018). In northwestern India, extensive residue burning during October to November is a recurring phenomenon and has been widely examined from multiple perspectives. Previous studies report that these burning events contribute to severe air-quality degradation in downwind urban centers (Singh et al., 2018; Jethva et al., 2019), alter aerosol loading and chemistry (Mhawish et al., 2022), modify aerosol vertical stratification and radiative forcing (Hsu et al., 2003; Vinjamuri et al., 2020; Banerjee et al., 2021), induce adverse health effects (Singh et al., 2021), and may influence regional hydrological processes (Kant et al., 2023). However, limited attention has been paid to investigate its effect on urban climate, especially on modulating lower atmospheric thermal budget which has been otherwise strongly evident in case of forest fire (Liu et al., 2018, 2019).

Across the northwestern India, dual cropping pattern including rice and wheat crop is predominately practised over roughly 4.1 million ha of land (NAAS, 2017). Such a cropping pattern leads to generation of huge crop residues that are low in nutrient content and rich in silica and ash. Typically, residues from rice-wheat cropping system possess limited economic value, as they are unsuitable for use as alternative fodder, bioenergy feedstock or as raw material in pulp and paper industry (Shyamsundar et al., 2019; Lan et al., 2022). Besides, with the introduction of mechanical harvester in the 1980s and enactment of groundwater preservation act in the late 2000s, in situ burning of agricultural residues has become a recurrent practice among the local farmers. This practice serves to expedite field clearance and reduce the turnaround period between rice harvest and the subsequent sowing of the

wheat crop (Balwinder-Singh et al., 2019). India produces an estimated 500 million metric tonnes (MT) of crop residues annually, of which 20–25% are disposed of through open-field burning. Crop residue burning is particularly prevalent in northwestern India, where roughly 20-25 MT of residues are set on fire each year (Balwinder-Singh et al., 2019; Lan et al., 2022). Unregulated residue burning in this region contributes approximately 300 Gg/yr of $PM_{2.5}$ and 50 Tg of $CO_2$ equivalent green-house gas emission (Singh et al., 2020). Notably, the frequency of fire incidences has exhibited a persistent upward trend, coinciding with concurrent increases in vegetation indices and atmospheric aerosol loading (Vadrevu et al., 2019; Jethva et al., 2019). In addition to atmospheric emissions, fires exert numerous biophysical impacts on the surrounding ecosystems. Fire induces a cascade of consequential processes, including modifications to the surface energy balance, redistribution of nutrients, alterations in species composition, changes in surface albedo, and variations in evapotranspiration rate (Ward et al., 2012; Liu et al., 2019). Additionally, fire can induce certain biogeochemical and biophysical stresses on local environment by modifying atmospheric composition and surface properties (Andela et al., 2017; Aditi et al., 2025). Such transformation of the native landscape, coupled with excessive release of energy, aerosols and its precursors, may therefore have several potential implications on the environment.

Most studies on biomass-based fires have focused on identifying land–atmosphere processes responsible for fire initiation and propagation, quantifying emissions, and evaluating fire-induced land–atmosphere exchanges (Lasko and Vadrevu, 2018; Jethva et al., 2019; Chuvieco et al., 2021; Aditi et al., 2025). In contrast, there is a paucity of knowledge regarding how biomass burning contributes to climate feedbacks through modifications of Earth's surface radiative budget and land surface temperature (Bowman et al., 2009; Andela et al., 2017). Plausible explanation to this includes limited observation and associated uncertainties in estimating key biophysical parameter like surface albedo, land-atmosphere exchange of sensible heat flux and water vapor, changes in evapotranspiration before and after fire events. There are instances when global forest fire incidences and size have been linked with modifications in land surface temperature (LST; Alkama and Cescatti, 2016; Liu et al., 2018, 2019). Likewise, Liu et al. (2019) noted an enhancement in mean annual LST over burned forest area in the northern high latitudes. Similar evidence of increase in summertime surface radiometric temperature over temperate and boreal forests in the Northern

Hemisphere was accounted by Zhao et al. (2024). Alkama and Cescatti (2016) reported
increases in mean and maximum air temperature over arid regions following forest loss,
highlighting the sensitivity of surface temperature to land-cover modification. However, fire-
induced thermal forcing is strongly constrained by the fire size (Zhao et al., 2024). Small, short-
lived fires, such as those associated with agricultural residue burning, often fail to produce
sufficiently large changes in surface albedo or evapotranspiration, and therefore may not
generate a detectable LST response. Incidence of elevated LST over different provinces in
China due to agricultural residue burning has only recently reported by Zhang *et al*. (2020). A
spatially heterogeneous increase in LST correlated strongly with fire count, with highest LST
gradient noted at distances of 4–10 km from the central point of crop residue burning and
persisting for 1-3 days. In contrast, the effects of post-harvest fire incidences in northwestern
India on LST remain largely unexplored. This gap introduces considerable uncertainty in
assessing the climate feedback of crop residue burning and highlights the need for a better
understanding of the underlying mechanisms.
This study aims to explore immediate biophysical effect of agricultural residue fire on
surface temperature over northwestern India. By integrating spatially and temporally
consistent satellite observations and reanalysis datasets, including fire counts, fire radiative
power, land surface temperature, aerosols, meteorological covariates, topography, surface
property, and physical environment over intensive fire zone, we sought to quantify time-
bound changes in LST in response to variations in fire intensity and aerosol loading. Several
statistical methods were applied to construct the changes in LST with fire severity and
aerosols. Additionally, a space-for-time framework was followed to assess the effects of
recurrent FRP variations on LST and aerosol optical depth (AOD) throughout the fire season.
Specifically, we addressed two key questions: (1) Does LST respond to changes in fire intensity
over northwestern India? and (2) How do local meteorology and aerosol loading modulate
LST variation with respect to space and time? To the best of our knowledge, this is the first
systematic assessment of agricultural residue fire–driven modulations in LST over
northwestern India. By integrating multiple geospatial observations, the analysis offers critical
insights into the biophysical feedbacks of residue-based fire and advances understanding of
LST responses to residue burning. Further, it refines estimates of fire-induced perturbations
in the regional radiative budget offering valuable representation of biomass-based fire in
Earth system models.

## 2. Dataset and methodology

### 2.1 Study domain

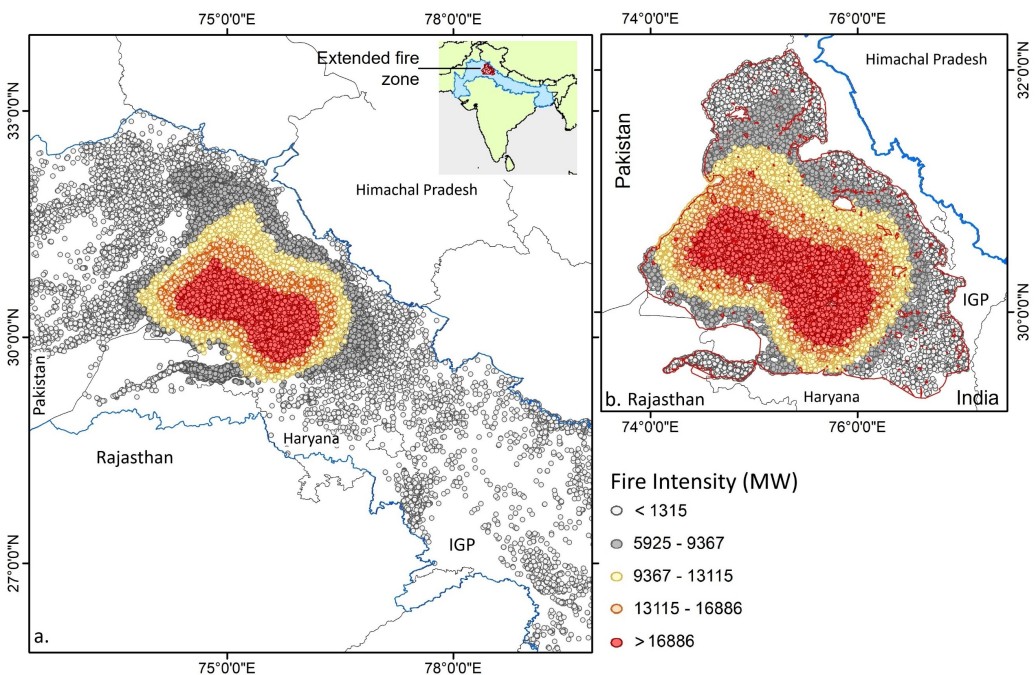

Fig. 1. Spatial variation in satellite-based fire radiative power across northwest India,
distribution of FRP-based fire intensity (MW/pixel) (a) and domain selected for
retrieval and processing of SNPP VIIRS FRP, AOD and Aqua MODIS LST (b). The region
marked with blue in Fig. 1a subset indicates the Indo-Gangetic Plain (IGP) spanning
from Pakistan to Bangladesh through India. The extended fire zone selected for
analysis is marked with red within the IGP and has been shown in detail in Fig. 1a with
fire pixel density. *Disclaimer note:* The international boundaries shown on this map are
for illustrative purposes based on the '*Survey of India'* archive. They do not imply
endorsement of acceptance by the journal or publisher of any particular political or
legal status of the territories depicted.
Post-harvest biomass burning is predominantly practiced across the northwestern
Indo-Gangetic Plain (IGP) of South Asia, particularly in the agrarian states of Punjab and
Haryana, which together contribute nearly 60–70% of India's total food grain production. The
concurrent rise in rice and wheat cultivation has led to a substantial increase in crop residue
generation, resulting in higher fire intensity in recent years (Jethva et al., 2019). In this study,
geospatial analyses of LST, fire activity, and aerosol loading were conducted over
northwestern India during October–November between 2017 and 2021. The combination of
high agricultural output, extensive biomass burning, and increasing fire activity makes this
region particularly suitable for investigating fire dynamics and their environmental
implications. Schematic workflow indicating core datasets and adopted methodology for
exploring FRP-AOD-LST association is illustrated in Fig. S1. Instead of defining a fixed spatial
domain a priori, year-wise fire signals were retrieved across cropland areas in northwestern
India. This approach allowed the delineation of a core study region that varied annually
according to year-specific fire intensity and spatial trends (as shown in Fig. S2), but all
eventually bound to 29.2770° to 32.1625° N and 73.8996° to 77.0718° E, as illustrated in Fig.
1b.
**2.2 Spatial dataset**
Active fire count data was retrieved from the standard fire product of Visible Infrared
Imaging Radiometer Suite (VIIRS) Collection-2 (VNP14IMG) available at 6-min L2 swath at 375
m resolution. The VIIRS onboard the Suomi National Polar-orbiting Partnership (SNPP) satellite
is a cross-track single-angle scanning radiometer which was launched in year 2011 under joint
operation of NASA and NOAA. The VIIRS fire detection algorithm typically extends well refined
and validated MODIS Fire and Thermal Anomalies product (Giglio et al., 2003). The I-band
based fire detection algorithm primarily utilizes brightness temperature of Channel I4 on
middle infrared spanning from 3.55 to 3.93 µm, centred at 3.74 µm. Additionally, to isolate
the active fire spots from the fire-free background channel, a single gain I5 at thermal infrared
regions (10.5–12.4 µm) is also considered. Rest of the I-band channels i.e. I1 to I3, covering
visible, near and short-wave IR are used to distinguish pixels with cloud, water and sun-glint
(Schroeder et al., 2014). The VIIRS fire database was considered due to its superior precision
and accuracy in identifying relatively small fire, greater spatial resolution at footprint and pixel
saturation temperature (Li et al., 2018; Vadrevu and Lasko, 2018; Aditi et al., 2023). For this
experiment, SNPP VIIRS 375 m L2 active fire count data with nominal (fire mask class 8) and
high confidence (fire mask class 9), was retrieved over northwestern India from year 2017 to
2021 (all inclusive).

Fire radiative power (FRP) quantifies the release of radiative energy from biomass

burning integrated at all angles and wavelengths over a spatial scale. Measured in Watt, FRP

retrieval quantifies the release of heat energy against time and in many instances linearly

associated with the rate of fuel consumption and emission (Ichoku et al., 2008; Nguyen and

Wooster, 2020). A detailed description on FRP retrieval and comparison among the sensors

are available in Wooster et al. (2003, 2005) and Ichoku et al. (2008). Li et al. (2018) concluded

VIIRS FRP as comparable with MODIS FRP in most of fire clusters and stable across swath.

Here, FRP (MW) was processed from the SNPP VIIRS C2 Level-2 (L2) 375 m active fire product

(VNP14IMG). VIIRS FRP was used as a proxy of fire intensity and potential emission strength

from the biomass burning area, and considered as a direct measurement of radiative energy

being released from individual fire pixel.

Land surface temperature (LST, in °C) at 1 km spatial resolution was utilized from

Moderate Resolution Imaging Spectroradiometer (MODIS) version 6.1 Land Surface

Temperature and Emissivity retrievals product (MYD11A1). Typically, LST indicates

thermodynamic temperature of the interface atmospheric layer within soil, plant cover and

lower atmosphere, and serves as an indicator of land-atmosphere interaction and exchange

(Li et al., 2023). Here, MODIS MYD11A1 radiometric dataset with quality flag '00' was

specifically chosen considering its broad swath and wider applicability in estimating land

surface temperature. MODIS LST is validated against ground observations on diverse land

covers and reported to provide realistic estimate of surface temperature (Wan, 2014) with an

uncertainty of ≤0.5 K. The dataset includes daytime maximum LST (at 1:30 PM local time) and

nighttime minimum LST (at 1:30 AM local time). Here, daytime LST dataset were obtained

solely from the MODIS sensor onboard the Aqua satellite to closely coincide with VIIRS fire

count observations at 1:30 PM local time, a period when crop residue–based fires are

expected to reach at peak.

Aerosol optical depth (AOD) from Visible Infrared Imaging Radiometer Suite (VIIRS)

sensor on-board SNPP satellite offers accurate estimation of columnar aerosol loading at 550

203        nm over land. Accuracy of VIIRS V1 DB AOD was evaluated extensively over South Asia by Aditi

et al. (2023) and reported to provide stable AOD retrieval against AERONET. Sayer et al. (2019)

reported an estimated error of ±(0.05+20%) in VIIRS Version 1 DB AOD dataset. Here, Deep

Blue (DB) Version 1 AOD dataset (AERDB_L2_VIIRS_SNPP Level-2) was used to retrieve AOD

with a nominal spatial resolution of 6 km at nadir. Only quality assured AOD (QA ≥ 2) was
retrieved for the months of October to November over selected spatial domain.
Terra/Aqua MODIS land cover data was used to discriminate crop land against the rest
to filter out thermal anomalies exclusively over the agriculture land. To achieve this, MODIS
L3 V6.1 Global Land Cover type product (MCD12Q1) was retrieved from LAADS DAAC site for
year 2017, available at 0.5 km spatial resolution. MODIS land cover types adopts International
Geosphere-Biosphere Programme (IGBP) and other land type classification schemes to
classify land cover. Here, land cover type 12 (cropland) was earmarked to isolate the
agriculture land from its surrounding (Fig. S3).
Daily composite data on surface and root-zone soil moisture (SM, m³ m⁻³) available at
9 km resolution was obtained from NASA's Soil Moisture Active Passive (SMAP) satellite
mission having L-band radar. The Normalized Difference Vegetation Index (NDVI) at 6 km
resolution was derived from the VIIRS/SNPP Deep Blue (AERDB_L2_TOA_NDVI) dataset and
was utilized to quantify surface vegetation greenness dynamics. Elevation data at 30 m
resolution was retrieved from Copernicus DEM - Global and European Digital Elevation Model
dataset for year 2015. Surface albedo data was acquired from MCD43 suite of NASA standard
product which integrates both Terra and Aqua retrievals. Here, white-sky version 6.1
shortwave albedo data (MCD43A3, Albedo_WSA_shortwave) at 500 m pixel resolution with
daily-time step (quality score: 0) was used.
Lower surface meteorological data including air temperature (AT), total solar radiation
flux (SR), precipitation (PR), relative humidity (RH) was procured from European Centre for
Medium-Range Weather Forecasts (ECMWF) AgERA5 dataset. The AgERA5 dataset has been
generated by Copernicus Climate Change Service (2020) from hourly ECMWF ERA5 dataset for
specific agro-ecological based applications. The meteorological data were pre-customized
with temporal aggregation aligned to local time zones and spatial enhancement to a 0.1°
resolution using grid-based variable-specific regression model. Here, air temperature at 2
meters above the surface, total solar radiation flux received at the surface over a 24-hour time
period, and relative humidity at 2 m height was selectively used over pre-identified intensive
crop-based fire zone. Planetary boundary layer height (PBLH) data at 0.25° x 0.25° resolution
was acquired from ECMWF ERA5 for 13:00-14:00 h local time corresponding with VIIRS
overpass time. A description of all core datasets used in this analysis and their resolution,
version, and quality flags is included in Table S1 (in supplementary file).

**2.3 Spatial analysis for fire-aerosols-LST association**

**2.3.1 Selection of intensive fire zone**

Post-harvest residue burning typically begins in mid-October and reaches peak
intensity by mid-November across northwestern India. Accordingly, all spatial analyses were
conducted for October and November for the years 2017–2021. The VIIRS 375 m fire product
successfully retrieved active fire pixels across the Indo-Gangetic Plain, capturing substantial
spatial heterogeneity. To ascertain a representative region having predominance of residue-
based fire, spatial comparison of fire pixel density was made using daily retrieved VIIRS FRP
dataset. FRP was selected instead of fire counts because it directly quantifies the radiative
energy released from active burning and therefore provides a more meaningful metric for
assessing potential impact on LST. FRP density was computed on a $1.5 \times 1.5$ km$^2$ grid to
characterize spatial variations in fire intensity across northwestern India. Following Giglio et
al. (2006), FRP density was estimated as the ratio of total FRP within a grid cell to the grid
area.

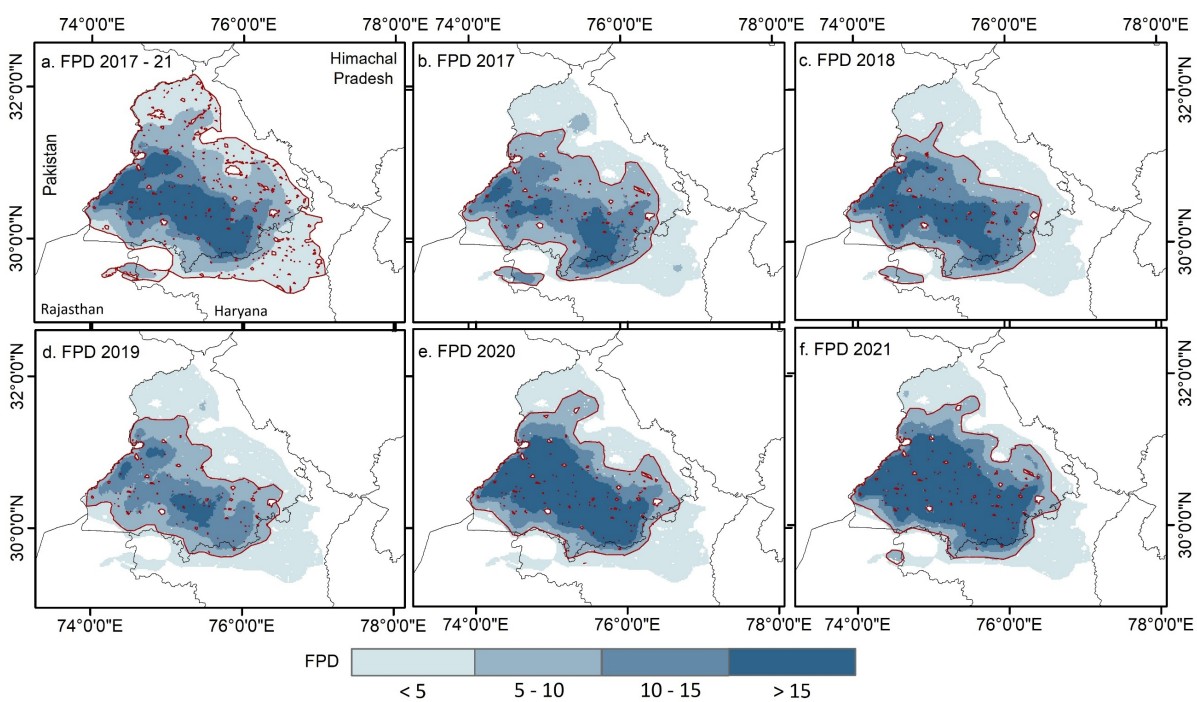


Fig. 2. Selection of high intensity residue-based fire zone based on fire radiative power pixel
density (MW 2.25 km$^{-2}$ day$^{-1}$). Fig. 2a indicates the '*extended geographical region*'

demarcating the entire area with varying fire intensity selected for spatial analysis. Rest

of the figures classify year-specific '*intensive fire zone*' based on FRP density.


Initially, geospatial variations in fire intensity and the associated changes in LST and

AOD were evaluated. Spatial intercomparison between FRP, LST, and AOD was performed
over the region delineated in Fig. 2a. This area was selected to encompass an extended
geographical domain without imposing thresholds on low or high FRP density across
northwestern India. The region is hereafter referred to as the "extended geographical
region," as it integrates fire activity across all years and was used exclusively to establish the
spatial association between the predictor (FRP) and dependent variables (LST and AOD).

In contrast, to assess the day-to-day influence of fire intensity and aerosol loading on

LST, a comparatively high-intensity fire zone was delineated relative to low-intensity areas.
To achieve this, the entire crop-residue burning region of northwestern India was mapped
using a constraint from low FRP density ($<5$ MW grid$^{-1}$) to high FRP density ($>15$ MW grid$^{-1}$).
Spatial variations in FRP density were evaluated for each year, and regions with FRP density
$>5$ MW grid$^{-1}$ were identified as the "intensive fire zone" (Fig. 2b–f). This threshold ensured a
better representation of the effect of medium to large crop-based fire on regional LST as
small-intensity fire deem to extinguish faster while being inconducive to considerably
influence surface temperature (Zhao et al., 2024).

All subsequent spatial datasets used for evaluating FRP–AOD–LST relationships were

retrieved exclusively within the year-specific 'intensive fire zone' having FRP density $>5$ MW
grid$^{-1}$. Notably, the spatial extent of the high-FRP region remained largely consistent across all
years (Fig. 2b–f), with areal estimates summarized in Table S2. It is noteworthy, the region was
pre-filtered based on the Terra/Aqua MODIS land cover data to deselect any FRP pixel that
emerged from a non-agricultural/crop land.
**2.3.2 Selection of temporal window**

After isolating the region with higher fire pixel density, the next step was to identify

the temporal window in which potential associations between fire intensity and other
explanatory variables could be examined. The temporal selection was based on two scenarios,
as illustrated in Fig. 3. Scenario 1 was designed to quantify the influence of FRP, aerosols, and
other parameters on LST during the period when fire activity begins to intensify and remains
persistent over the intensive fire zone. Scenario 1 defines the initiation day as the first instance
in October when aggregate FRP consistently exceeds 1500 MW and shows at least a 50%
increase compared to the previous day. The scenario concludes in November when aggregate
FRP decreases by at least 50% relative to the previous day. The selected dates for Scenario 1
are listed in Table S3, with two exceptions. First, in year 2018 when a >50% criteria was not
met despite having an aggregate FRP >1500 MW and second, in year 2017 when a prior
decrease (>50%) in FRP was avoided because of subsequent rise in fire intensity.

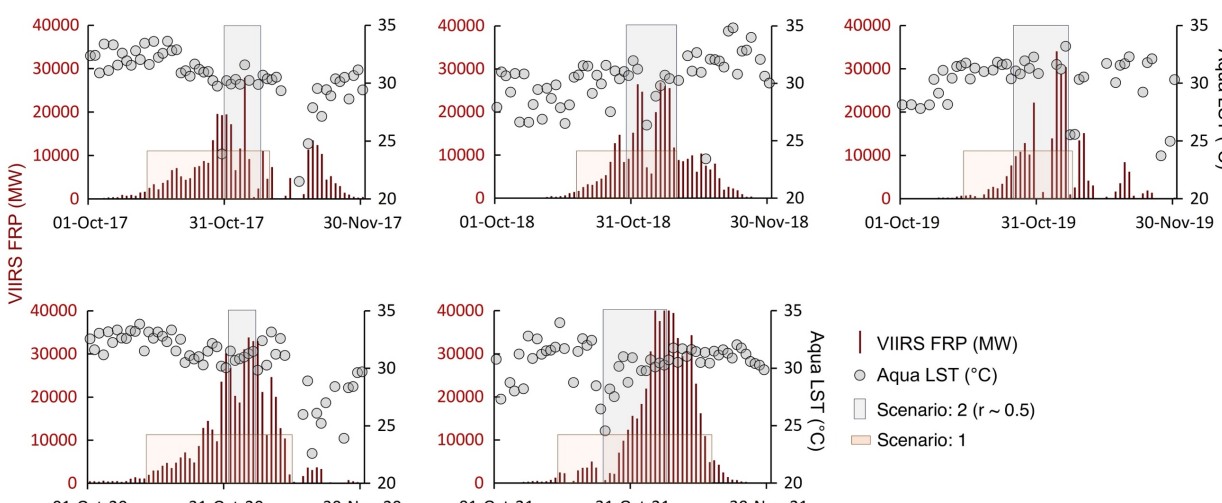


Fig. 3. FRP and LST time series over intensive fire zone showing the extent of scenarios used
for geospatial modelling.
To define Scenario 2, a statistical association was examined between day-specific
aggregate FRP and the spatially averaged LST. Pixel-based LST values were averaged over the
intensive fire zone and compared against the area-weighted sum of FRP on a day-to-day basis.
A temporal window ("Scenario 2" in Fig. 3) was selected using two criteria: (i) the end of the
window had to coincide with a period of persistently high FRP, and (ii) the window had to
exhibit a strong positive correlation (r ≥ 0.5) between FRP and regional LST. Such restricted
criteria were put to ensure that we only select year-specific window(s) when FRP (so the fire
count) increases with time and exhibit a strong association with regional LST. Descriptive
statistics of both scenarios are included in Table S4. It is noteworthy that selecting multiple
windows within a year having coinciding days was avoided while ensuring windows should not
contain more than 5% of missing days, irrespective of parameters.

**2.4 Spatial correlation between fire, aerosols and LST**

To examine the spatial association among FRP, LST, and AOD over the residue–based fire zone, grid-based spatial correlation coefficients were computed, and their statistical significance ($p < 0.05$) was tested across the study domain. Daily FRP (375 m) and LST (1 km) datasets were initially resampled to a 6x6 km$^2$ resolution to match the VIIRS AOD dataset before subject to spatial correlation analyses among the predictor and dependent variables. This approach facilitated the identification of regions exhibiting strong co-variability in thermal conditions corresponding to variations in fire intensity and columnar aerosol loading.

**2.5 Hurst Exponent**

The Hurst exponent is a statistical measure used to characterize the properties of a time series without imposing assumptions about its underlying distribution. Originally introduced by Hurst (1951) in hydrological studies and later refined by Markonis and Koutsoyiannis (2016), it has since been widely applied across diverse scientific disciplines to analyse long-term trends and variability. In this study, the Hurst exponent was computed for FRP, AOD, and LST time series to identify long-term statistical persistence in the datasets. To estimate the Hurst exponent at the spatial scale, $6 \times 6$ km$^2$ resampled datasets of FRP, AOD, and LST were used. Adjustment of seasonal cycle was not accounted, as the datasets were retrieved and processed exclusively for a single season across the selected years. The main calculation procedures were as follows (Granero et al., 2008):

A time series x(t) is given,

$$(x)_t = 1/\tau \sum_{t=1}^{\tau} x(t) \quad t = 1, 2, 3 \dots \tag{1}$$

The cumulative deviation is determined using Eq. 2:

$$X(t, \tau) = \sum_{u=1}^{\tau} (x(u) - (x)_t), \text{ with a condition of } 1 \leq t \leq \tau. \tag{2}$$

Extreme deviation sequence, is defined as:

$$R(\tau) = \max_{1 \leq t \leq \tau} X(t, \tau) - \min_{1 \leq t \leq \tau} X(t, \tau) \ where \ \tau = 1, 2, 3 \dots \tag{3}$$

The standard deviation sequence is calculated by Eq. (4):

$$S(\tau) = [1/\tau \sum_{t=1}^{\tau} (x(t) - (X)_\tau)^2]^{1/2} \ where \ \tau = 1, 2, 3 \dots \tag{4}$$

By considering both extreme deviation sequence and standard deviation sequence,

R/S = R ($\tau$ )/S ($\tau$) when assuming (R/S) $\propto$ ($\tau$/2) $^{H}$                                    (5)

The Hurst exponent ranges between 0 and 1. A value of 0.5 indicates that the time

series behaves as a purely stochastic process without persistence, implying that future
variations are independent of past behaviour. Values greater than 0.5 denote statistical
persistence, reflecting a tendency for future changes to follow the same trend as in the past,
with higher values corresponding to stronger persistence. Conversely, values below 0.5
indicate anti-persistence, suggesting a tendency for the time series to reverse its trend over
time; lower values represent stronger anti-persistence (Peng et al., 2011).
**2.6 Space-for-time approach**

A space-for-time approach was employed to assess and compare the changes in LST

and AOD with respect to FRP within the extended geographical region experiencing recurrent
medium- to high-intensity fire. To ensure that changes in LST and AOD were attributable solely
to fire activity, grids with similar characteristics in terms of topography, climate, and physical
environment were compared (Liu et al., 2019). To achieve this, daily datasets including
meteorological covariates (PBLH, AT, SR, RH and PR), physical environment (elevation),
vegetation and soil characteristics (NDVI, soil moisture), climatological mean LST and AOD,
and surface property (albedo) were extracted over both fire and no-fire grids at a spatial
resolution of 10 × 10 km². The daily data were retrieved for each grid under Scenario 2, when
FRP reached its peak and exhibited a positive association with regional LST.

After filtering out the grid cells with missing LST or AOD values, remaining grids were

classified into two groups: those with zero FRP (no-fire) against the grids having FRP > 0,
indicating presence of fire. Fire and no-fire grids with comparable spatial characteristics were
grouped into a single stratum, and a stratified matching technique was applied to generate
multiple strata based on combinations of the selected confounders. Grids were retained only
when differences in their physical environment, vegetation and soil characteristics, climate
and land cover between fire and no-fire conditions were smaller than the defined thresholds
($\Delta$elevation < 50 m; $\Delta$NDVI <0.05; $\Delta$soil moisture <0.05; $\Delta$albedo <0.05; $\Delta$LST <10.0; $\Delta$AOD
<0.80). Comparisons were then made within strata containing grids of similar attributes to
ensure that the observed variations in LST and AOD could be attributed solely to fire activity.
The difference in LST ($\Delta$LST) among the fire grids (LST$_{fire}$) and grids exhibiting no-fire (LST$_{no-fire}$)
having similar attributes were compared to constitute effect of residue-based fire on LST. A
positive (negative) $\Delta$LST (LST$_{fire}$ – LST$_{no\text{-}fire}$) indicates fire-induced warming (cooling) and was
used to quantify changes in LST associated with residue burning for the selected years. A
similar approach was also adopted to evaluate $\Delta$AOD variations using grid-based retrievals.
It is noteworthy that the grids were not classified based on meteorological covariates,
as only insignificant variations were noted among the grids. The entire northwestern cropland
experiences a relatively uniform background climate during October–November, including
comparable boundary layer heights, with PBLH standard deviations ranging from ±10 m to
±33 m within a single fire season. The climatological mean LST and AOD were computed only
for the pre-fire season (September, 2017-2021), during which none of the grids experienced
residue-burning activity. Furthermore, grids were not differentiated by slope or aspect, given
the minimal topographic variation across the Gangetic Plain.
**2.7 Multicollinearity assessment**
Multicollinearity, where independent variables are highly correlated, can distort
regression estimates and obscure the true contribution of individual predictors (Graham,
2003). To assess this, the Variance Inflation Factor (VIF) for all covariates was calculated using
the *statsmodels* library. A VIF of 1 indicates no correlation, values between 1 and 5 suggest
moderate correlation, and values greater than 5 are generally interpreted as evidence of
substantial multicollinearity (Daoud, 2017). All biophysical, land-surface, and meteorological
variables met acceptable VIF thresholds, except solar radiation, which was therefore excluded
from Random Forest and GWR analysis. Additionally, soil moisture data was removed from
further analysis due to a high percentage of missing observations (~30%).
**2.8 Random Forest regression**
Random Forest regression was used to model the relationship between the
dependent variable (LST) and predictor variables (AOD, PBLH, AT, RH, SR, PR, NDVI, elevation,
albedo, and FRP) within the intensive fire zone. Daily retrievals, averaged over the year-
specific intensive fire area, were incorporated into the ensemble framework to capture
potential non-linear associations among variables. The selected approach ensures robustness
to multicollinearity, minimizes overfitting, and effectively captures complex predictor
interactions.
Random Forest is a non-linear ensemble machine learning algorithm that constructs
multiple decision trees from bootstrapped samples of the training data, with a random subset
of predictors evaluated at each split. Final predictions are obtained by averaging all trees,
improving generalization and reducing overfitting (Breiman, 2001; Puissant et al., 2014). The
algorithm was selected due to its strong predictive capability, scalability to large
environmental datasets, resilience to correlated inputs, and demonstrated success in
previous LST-related studies (Logan et al., 2020; Wang et al., 2022; Zhang et al., 2025). These
attributes collectively support Random Forest as an appropriate and interpretable choice for
assessing the complex interactions between fire intensity, aerosol loading, and LST dynamics.
Key Random Forest hyperparameters (n_estimators, max_depth, min_samples_split,
min_samples_leaf, and max_features) were optimized using Bayesian optimization
implemented via BayesSearchCV in *scikit-optimize* (Snoek et al., 2012; Shahriari et al., 2015;
Frazier, 2018). This adaptive, probabilistic search strategy efficiently identifies near-optimal
hyperparameter combinations while minimizing computational cost. To ensure robust model
evaluation and mitigate temporal dependence, we employed temporal block cross-validation
using a 3-fold GroupKFold in the *scikit-learn* library, where all observations from a given year
were assigned to the same fold. This approach prevented temporal overlap between training
and validation datasets and reduced information leakage across years. This approach also
minimized temporal autocorrelation and prevented data leakage across time periods. Model
performance was quantified using cross-validated coefficient of determination ($R^2$), Root
Mean Squared Error (RMSE), and Mean Absolute Error (MAE), providing a comprehensive
assessment of model accuracy and prediction error.
**2.9 Assessment of relative feature importance**
Variable importance was derived from the trained RF model using the mean decrease
in impurity method, which quantifies each predictor's relative contribution to reducing
variance in model predictions. This approach provides insight into the dominant factors
governing the spatial and temporal variability of LST. Feature importance values were
extracted and ranked to identify the most influential predictors under different fire intensity
scenarios. To enable direct comparison among predictors, the relative contribution of each
feature was expressed as its importance score normalized by the sum of all feature
importances. As Scikit-learn's RandomForestRegressor.feature_importances_ inherently
returns normalized values summing to one, the reported scores directly represent each
predictor's proportional influence within the model.

**2.10 Spatial heterogeneity assessment using GWR**

Spatial heterogeneity in the influence of FRP, AOD, and other spatial predictors on LST
within the intensive fire zone was assessed using Geographically weighted regression (GWR)
at 1x1 km$^2$ grid. GWR is a spatially explicit regression technique designed to quantify how
relationships between predictors and a dependent variable vary across geographic space by
estimating spatially varying coefficients (Brunsdon et al., 1996). The method applies a
distance-based weighting scheme, whereby observations closer to a given location receive
higher weights, allowing local parameter estimation that reflects neighbourhood-specific
dynamics (Yang et al., 2020). Unlike global regression models that assume spatial stationarity,
GWR produces location-specific coefficient estimates, offering a more nuanced
understanding of spatially varying associations between LST and its predictors (Fotheringham
et al., 2009). The GWR model is formally expressed as:
$$y_i = \beta_0(ui, vi) + \sum_{k=1}^{m}(\beta k(ui, vi)\, xik) + \varepsilon i \qquad (6)$$
where (ui, vi) are the coordinates of observation i, βk(ui, vi) are spatially varying coefficients,
xik are predictor variables, and εi denotes random error. In GWR, local parameters are
estimated using weighted least squares, where each observation is assigned a weight based
on its spatial proximity to the location being evaluated. These weights are determined by a
spatial kernel function and a bandwidth parameter that defines the extent of spatial
influence. Selecting an optimal bandwidth is therefore essential to balance the trade-off
between model bias and variance. In this study, the optimal bandwidth was identified through
an iterative optimization procedure that minimizes the corrected Akaike Information
Criterion (AICc) (Fotheringham et al., 2009). This approach ensures robust estimation of local
relationships while effectively accounting for spatial non-stationarity in the dataset. Such a
framework is particularly valuable in fire-affected landscapes, where the impacts of fire
intensity, aerosol loading, and surface characteristics on LST are inherently heterogeneous
and vary substantially across space.


## 3. Results and discussions

### 3.1 Spatial association between fire, aerosols and LST

Spatial variations in FRP, LST and AOD averaged for October to November between 2017 and 2021 over extended geographical region is shown in Figure 4(a-c). While residue-based FRP did not exhibit a distinct spatial pattern, temporal variations were prominent, with monthly mean FRP in November (310,188 MW month$^{-1}$) showing nearly a 100% increase compared to October (152,616 MW month$^{-1}$; Table S5). In contrast, the spatial pattern of LST exhibited considerable heterogeneity, with relatively higher temperature observed in the southern parts of the region that gradually declined northward. This north–south gradient may be partially attributed to the proximity of the Himalayan foothills, where the cooler mountainous environment likely offsets fire-induced surface warming. A gradual decline in spatially averaged monthly mean LST was also accounted in November (29.0±2.4 °C) compared to October (31.0±1.6 °C).  A spatially distinct pattern in columnar aerosol loading was evident across the extended geographical region, with elevated AOD (> 0.65) retrieved over the central areas that gradually decreased towards its periphery (< 0.30). Such spatial variability in aerosol loading is likely driven by differences in the intensity of residue-based fires and the associated emissions of aerosols and trace gas precursors. Moreover, the pronounced increase in monthly mean AOD (October: 0.59 ± 0.08; November: 0.82 ± 0.12) likely reflects the intensification of fire during early November, compounded by concurrent meteorological influences, most notably the seasonal decline in boundary layer height (Banerjee et al., 2022).

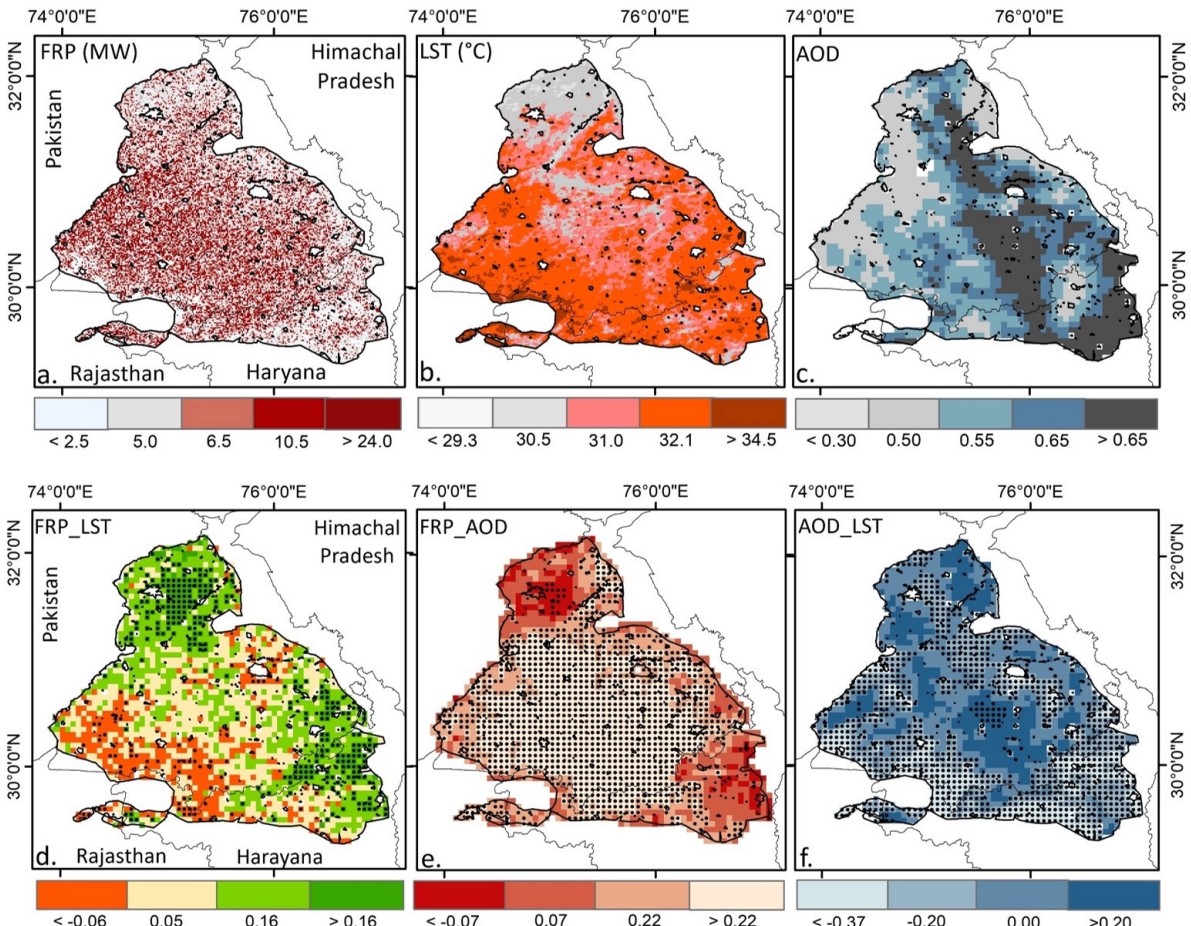

Fig. 4. Spatial variations of FRP, LST and AOD over extended geographical region, 5-year mean FRP (a), LST (b) and AOD (c), and spatial correlation between FRP_LST (d), FRP_AOD (e) and AOD_LST (f). To compute spatial correlation, daily retrievals of FRP, AOD and LST were converted to a common 6x6 km$^2$ grid. Spatial correlation was computed for the entire duration and significant correlation (P<0.05) is shown with black dot.

Spatial associations among VIIRS-derived FRP, MODIS LST, and VIIRS-based AOD daily retrievals were assessed over the extended geographical region (Fig. 4d–f). Spatial correlation between pixel-based FRP against LST reveals positive but spatially heterogenous association across most parts of the study area, except in the southern region. A statistically significant relationship (P < 0.05) between FRP and LST underscores the potential influence of crop residue burning on surface temperature. Similarly, a significant association between FRP and AOD was observed across the central region, where fire intensity was notably higher than in surrounding areas. This spatial covariation between fire intensity and columnar aerosol loading further reinforces the influence of biomass-burning-induced emissions of aerosols

and their precursors on atmospheric aerosol abundance. Biomass-burning aerosols,
predominantly composed of carbonaceous soot particles, are known to modulate the thermal
budget of the lower atmosphere (Freychet et al., 2019; Xu et al., 2021). The spatial association
between AOD and LST further supports the existence of a fire–aerosol–surface temperature
nexus over northwestern India. A comparatively weak yet statistically significant positive
correlation between AOD and LST likely reflects lower-atmospheric warming induced by
smoke aerosols, consistent with the similar warming effect over western United States during
2017 California wildfire (Gomez et al., 2024).
**3.2 Evaluation of Hurst exponent**
The Hurst exponent was evaluated to assess the long-term persistence of fire
intensity, surface temperature, and aerosol loading time series over the extended
geographical region. In principle, the Hurst exponent is used to quantitatively distinguish a
purely stochastic time series (H = 0.50) from a persistent (H > 0.50) or anti-persistent (H <
0.50) time series of pixel-based FRP, LST, and AOD, following the methodology described in
Markonis and Koutsoyiannis (2016) and Chen et al. (2022).
As shown in Figure 5, nearly the entire extended geographical region of northwestern
India exhibits Hurst exponent values greater than 0.50 for FRP, with relatively higher values
(0.60–0.70) concentrated toward its central zone. Although variations in Hurst exponent for
FRP was spatially inconsistent, primarily due to temporal and spatial fluctuations in fire
intensity, the FRP time series over most of the region indicates statistical persistence.
Similarly, elevated Hurst exponent values for LST (>0.50) across the region also exhibits
persistence at long run. Notably, the northern portion of the study region shows slightly
higher Hurst exponent values compared to the southern part. For regional aerosol loading,
except few isolated patches, comparatively high Hurst exponent values (>0.75) were
observed over the central region. Notably, this area also coincides with zones characterized
by high AOD (>0.65) and a statistically significant FRP–AOD association. Overall, the Hurst
exponent analysis indicates that the observed FRP, LST, and AOD time series across most of
the residue-burning region exhibit statistical persistence.

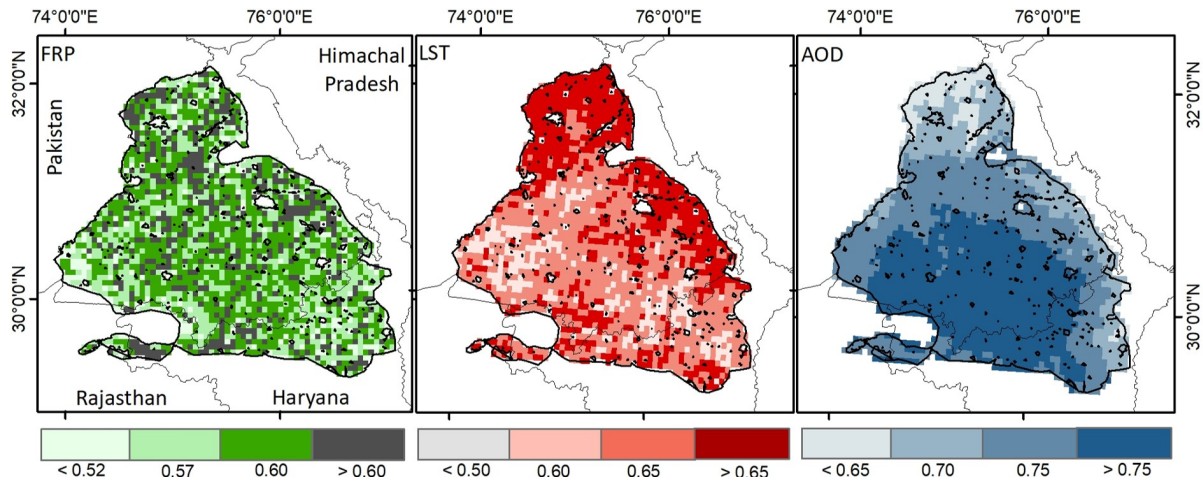

Fig. 5. Estimating FRP (MW), LST (°C) and AOD time-series persistence in extended geographical region.

However, interpretation of the Hurst exponent results should be approached with caution. The five-year dataset used here may not be sufficient to derive statistically robust estimates. For the same reason, trend analysis was not undertaken, as the limited dataset constrains the reliability of such estimates and falls beyond the scope of the present study. Nonetheless, several studies have documented long-term trends in fire dynamics and aerosol loading over northwestern India (e.g., Vadrevu and Lasko, 2018; Jethva et al., 2019; Singh et al., 2020).

### 3.3 Surface temperature and aerosols response to fire intensity

Fire intensity in terms of pixel-based FRP, aerosol loading and surface temperature were retrieved to compute corresponding daily and spatial means based on five years of satellite retrievals. It is noteworthy that to account immediate response of fire intensity and aerosol loading on surface temperature, all variables were retrieved exclusively over year-specific intensive fire zones, having cumulative FRP $\geq$ 5 MW grid$^{-1}$, as illustrated in Fig. 2(b-f).

A distinct temporal pattern is evident in the FRP time series (Fig. 6a), which corresponds closely with daily variations in fire counts (Fig. S4). Over northwestern India, FRP starts to build-up typically in mid-October, peaks consistently during the first week of November, and declines thereafter by mid-November. In contrast, the temporal pattern of the five-year mean LST time series appears less pronounced, as daily retrievals exhibit substantial variability. Regional LST demonstrates both interannual and intra-annual fluctuations, as illustrated in Fig. S5. Notably, the FRP time series aligns well with the mean columnar aerosol loading,

underscoring the potential influence of aerosol and precursor emissions from widespread
biomass burning. The characteristic rise in AOD during the first two weeks of November likely
represents a direct response to intensified fire activity, as columnar AOD values consistently
exceed 1.00 over the intensive fire zone. Interestingly, between October 25 and November
20 each year, approximately 90% of daily AOD observations surpass the five-year mean (0.74
± 0.28), coinciding with an 800% increase in average FRP (13,085 ± 6,825 MW) compared to
the remainder of the season (1,148 ± 1,478 MW). During this interval, the five-year mean
columnar AOD exhibits a strong association with the aggregate FRP (r = 0.46) and mean LST
(r = 0.41), whereas these associations weaken considerably outside this period (AOD–FRP: r =
0.18; AOD–LST: r = –0.02).

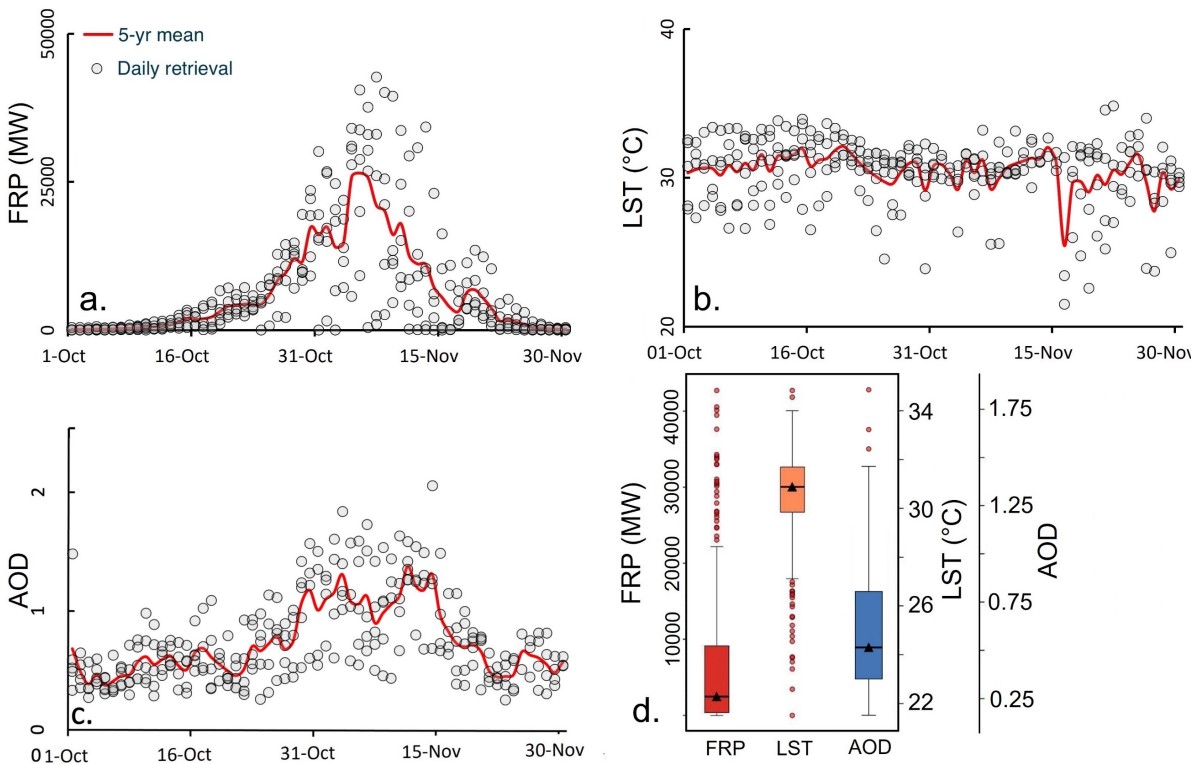


Fig. 6. Time series of five-year mean fire radiative power (FRP, a), land surface temperature
(LST, b) and aerosol optical depth (AOD, c) against daily retrievals, (d) covariation of FRP,
AOD and LST over intensive fire zone. Gray dots show daily retrievals from October to
November (2017–2021), with the red line depicting the corresponding 5-year mean.
The temporal associations among FRP, AOD, and LST clearly demonstrate the
immediate response of fire-induced variations in aerosol loading and surface temperature
over northwestern India. Accordingly, in the subsequent section, these relationships were
modelled using a geospatial tree-based regression framework that integrates concurrent
temporal features (e.g., day-specific retrievals) and spatial predictors (e.g., regional
meteorology, aerosol loading, and fire intensity) to quantify and characterize the FRP–AOD–
LST nexus within the intensive fire zone.

**3.4 Fire induced change in LST and AOD**

The effect of crop residue burning on land surface temperature and aerosol loading
was assessed using a space-for-time approach by overlaying grid-based VIIRS LST, FRP, and
AOD datasets over the northwestern region experiencing recurrent fire. To remove potential
confounding effect, fire and no-fire grids were retained for comparison only when they
matched in terms of topography, meteorology, physical environment, vegetation and soil
characteristics, climatological mean LST and AOD, and surface property. Comparisons were
performed within defined strata containing grids with identical characteristics to ensure that
the quantified changes in LST and AOD could be attributed solely to fire. A total of 7489 paired
no-fire and fire grids were used between 2017 and 2021 to quantify the relative change in LST
and AOD. It is noteworthy that all grids, whether exhibiting fire or not, were selected from
within the extended geographical region to capture localized variations in temperature and
aerosol loading.

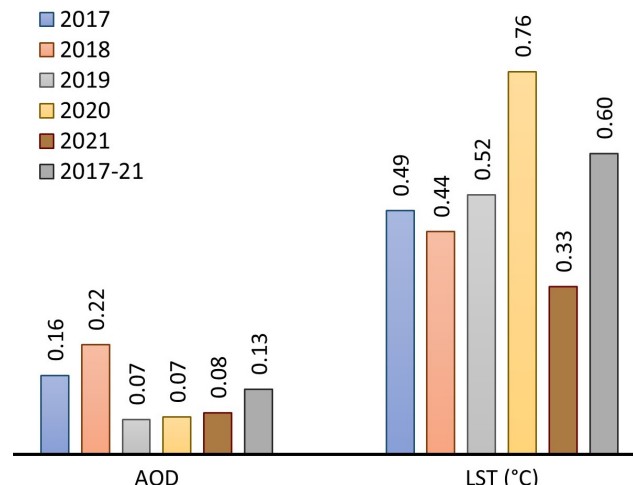

Fig. 7. Crop residue-based fire induced changes in land surface temperature and aerosol
loading.
As illustrated in Fig. 7, a consistent yet temporally dynamic increase in both LST and
AOD was observed over regions affected by residue-based burning compared with no-fire
zone. However, the magnitude of LST and AOD change across the fire zone was spatially
heterogeneous. On average, residue-based burning induced an increase of 0.60 °C in LST
during 2017–2021, with interannual variability ranging from 0.33 °C to 0.76 °C. This indicates
that residue burning exerts a persistent warming influence on land surface temperature, likely
driven by reduced evapotranspiration, enhanced shortwave absorption, increased sensible
heat flux, and fire-induced changes in surface albedo. However, a strong spatial heterogeneity
in LST and AOD modulation further indicates the potential influence of key confounding
factors and intensity of fire in regulating the change. The results of this study align with Liu et
al. (2019), who attributed a 0.15 °C rise in surface temperature over burned areas globally to
satellite-observed forest fires, as well as Liu et al. (2018), who documented a net warming
effect over the Siberian boreal forest. Additional evidence from Alkama and Cescatti (2016)
and Zhao et al. (2024) also indicates a positive linkage between forest fire occurrence, fire
intensity, and surface temperature. In contrast, the biophysical effects of agricultural residue
burning on land surface temperature remain poorly constrained. Zhang et al. (2020) reported
LST increases of 1–3 °C over three provinces in China associated with crop residue burning.
However, the feedback effects of meteorological covariates and systematic land-cover
differences on fire occurrence were not accounted for, leading to causal attribution of fire to
LST remains tentative.
A consistent annual increase in aerosol loading was also observed over the fire-
affected grids over northwestern India. A clear upward trend in AOD was noted across the
fire zones, with a mean increase of 0.13 AOD year$^{-1}$ and a range of 0.07–0.22 AOD year$^{-1}$. The
change in columnar aerosol loading, however, was spatially heterogeneous. Overall, the
increase in AOD from fire-associated emissions of aerosols and their gaseous precursors
reinforces the source-specific contribution of crop residue burning, a phenomenon well
documented in previous studies (Vinjamuri et al., 2020; Mhawish et al., 2022).
To quantify uncertainty in the estimated differences between fire-affected and non-
fire-affected grid cells, we further computed 95% confidence intervals for ΔLST and ΔAOD
using nonparametric bootstrapping. For each variable, 10,000 bootstrap samples were
generated by resampling grid cells with replacement, and the mean difference was
recalculated for each bootstrap replicate. The 2.5$^{th}$ and 97.5$^{th}$ percentiles of the resulting
sampling distribution were taken as the bounds of the 95% confidence interval (CI).
Nonparametric bootstrapping results into significant increase in both ΔLST (0.57°C; 95% CI:
0.33–0.81°C) and ΔAOD (0.13; 95% CI: 0.08–0.17) in fire-affected regions. Because both CIs
do not overlap zero, these differences are statistically robust and unlikely to be due to
sampling variability.

**3.5 Spatial regression of fire intensity and aerosols on LST**

A machine learning algorithm was employed to establish the statistical association
between the dependent variable LST and multiple predictors including fire radiative power,
aerosol loading, regional meteorology (Fig. S6), surface properties, and vegetation
characteristics. All biophysical parameters, except SR and soil moisture, retrieved under two
pre-defined scenarios, (one) days with moderate-to-high fire intensity and (two) days with
sustained high fire intensity exhibiting a positive association with regional mean LST, were
used to model the FRP–AOD–LST relation. Relative feature importance (RFI) of selected
predictors was first evaluated for the fire season, and the marginal effects of FRP and aerosols
on LST were subsequently quantified. Figure 8(a) presents the normalized RFI values for all
predictors under both scenarios, and the Random Forest hyperparameter tuning procedure
is summarized in Table S6. RFI quantifies the sensitivity of regional LST to each predictor and
reflects their partial contribution to surface temperature variability. Fire radiative power
emerged as the dominant predictor under both scenarios, indicating the strong influence of
fire-related energy release on regional radiative balance, likely through reduced
evapotranspiration and fire-induced changes in surface albedo (Liu et al., 2018, 2019).
Notably, the RFI was substantially higher during period of sustained high-intensity burning
(Scenario 2; RFI = 0.40) compared with days characterized by moderate-to-high fire activity
(Scenario 1; RFI = 0.22), highlighting the stronger thermal response associated with intensive
burning condition.
Next to FRP, PBLH exerted a significant influence on LST (RFI: 0.21–0.24), followed by
atmospheric temperature (RFI: 0.09–0.21). The strong effect of PBLH on LST can be explained
by restricted turbulent mixing during shallow boundary-layer conditions in post-monsoon
season. A relatively low PBLH (mean±SD: 71±29 m) over northwestern India reduces vertical
mixing and traps fire-induced heat and aerosols close to the surface (Vinjamuri et al., 2020).
This enhances shortwave absorption, suppresses evaporative cooling, and limits turbulent
heat dissipation, resulting in a stronger and more persistent increase in LST. Another notable
finding was the modification of LST due to enhanced columnar aerosol loading during fire
season. The RFI of AOD varies from 0.09 to 0.11, indicating its influence on regional radiative
budget. Residue burning releases aerosols and their gaseous precursors, which can exert
significant radiative impacts and drive rapid adjustments in both surface and atmospheric
temperature (Freychet et al., 2019; Xu et al., 2021). Fire-generated aerosols influence the
energy balance through scattering and absorption of radiation, alterations in cloud
microphysics, and changes in surface albedo via deposition of carbonaceous particles.
However, the magnitude and direction of these radiative effects remain uncertain at the
global scale (Tian et al., 2022). The partial influence of all other parameters, including
meteorological variables, land characteristics and elevation was less significant (RFI < 0.30).

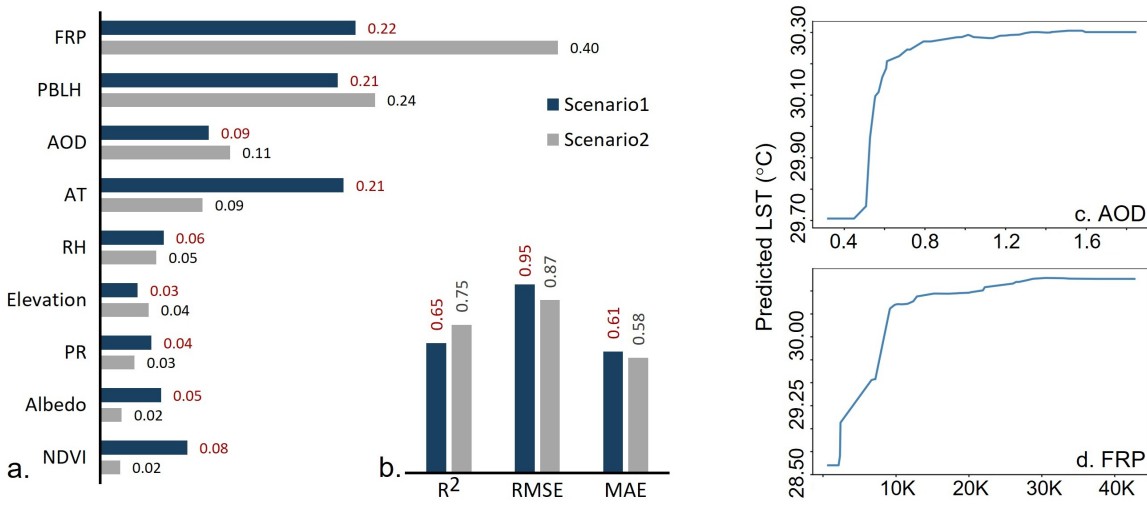


Fig. 8. Normalized relative feature importance of predictor variables on LST (a), cross-

validated evaluation of random forest performance (b), and partial dependence plots

of LST on AOD (c) and FRP (d). Here, K indicates x1000. The PDP plots are based on

scenario 2. Both RMSE and MAE have unit °C.


The predictive skill of the random forest model was assessed using temporal block

cross-validation to minimize temporal autocorrelation and prevent data leakage. Under both
scenarios model performance was found satisfactory with $R^2$ varying from 0.65-0.75, marked
with relatively low RMSE (0.87-0.95 °C) and MAE (0.58-0.61 °C). A satisfactory model
performance also ensures that residue burning provide a clear LST response and the RF model
was able to resolve non-linear land–atmosphere interactions, irrespective of the selected
scenarios. Relatively better performance was however, achieved in scenario 2 during the fire
days having better spatial association between FRP and LST. Collectively, this confirms that
moderate-to-high intensity residue burning leaves a measurable and predictable thermal
signature on the land surface over northwestern India.
The partial dependence plots (PDPs) in Fig. 8(c–d) illustrate the marginal effects of FRP
and AOD on LST. These plots show the expected change in LST associated with variation in
each predictor while holding all other predictors constant. The estimated effects of both FRP
and AOD exhibit a non-linear, saturating response. LST increases sharply at low-to-moderate
values of each predictor but the effect progressively weakens at higher magnitudes,
approaching an asymptotic limit. This behaviour likely arises from the complex interplay of
radiative and thermodynamic processes associated with biomass-burning emissions. Fire-
originated aerosols exert both direct and indirect radiative effects whose magnitudes and
signs vary with aerosol loading and composition (Freychet et al., 2019; Xu et al., 2021; Tian et
al., 2022). At moderate aerosol loading, UV-absorbing black carbon aerosols may enhance
atmospheric heating and can transiently increase near-surface temperature (Jacobson, 2001).
Fire-induced convective plumes may initially enhance surface temperatures, whereas strong
aerosol build-up can reduce solar transmittance to the ground. Aerosol–cloud interactions
further contribute to non-linearity by modifying cloud microphysics, lifetime, and albedo,
altering the regional radiative balance. Additionally, aerosol-driven changes in boundary-layer
structure, evapotranspiration, and soil moisture introduce additional land–atmosphere
feedbacks. Together, these interacting processes operate across multiple spatial and
temporal scales and do not scale linearly with aerosol loading or fire intensity, producing the
observed non-linear LST response. The RF model therefore provides strong evidence that
both fire intensity and fire-derived aerosols exert measurable and non-linear effects on
regional LST, with potentially important implications for the regional radiative budget.
**3.6 Geographically weighted regression on LST**
A Global Moran's I test was first applied to assess spatial autocorrelation in LST across
the intensive fire zone for the cumulative five-year period. As shown in Table S6, Moran's I
was 0.225, accompanied by a high positive Z-score and a statistically significant p-value (<
0.001), indicating a clustered spatial pattern of LST that is highly unlikely (<1%) to have arisen
by random chance. Given this spatial dependence, GWR was employed to evaluate spatial
heterogeneity in the relationships between LST, FRP, and other predictors. All variables used
in the Random Forest model were incorporated into the GWR framework under both pre-
defined scenarios. Model specifications and performance metrics including bandwidth and
kernel details are mentioned in Table S8.

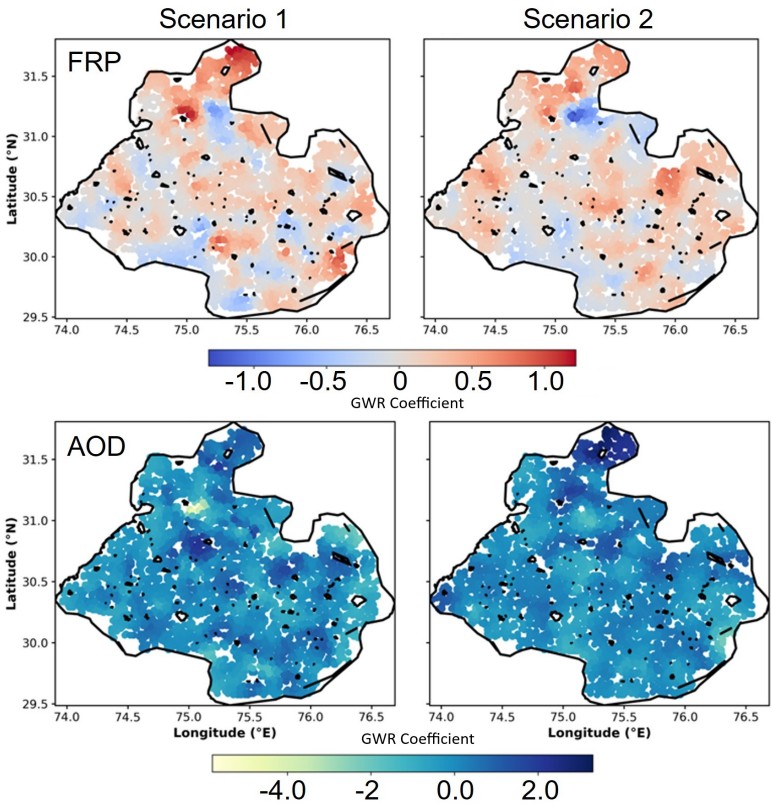

Fig. 9. Spatial distribution of FRP and AOD GWR coefficients across intensive fire zone.
GWR model demonstrated strong explanatory power, with global R² values exceeding
0.74, confirming that the selected predictors effectively captured spatial variability in LST. FRP
consistently showed a positive and spatially varying association with LST across both
scenarios, underscoring its dominant influence in fire-affected regions. Aerosol loading
demonstrated weak but spatially heterogeneous effects, reflecting localized differences in
aerosol–temperature interactions. Other predictors, including NDVI, RH, AT, PBLH, elevation,
and albedo (Fig. S7), exhibited local coefficients ranging from −0.76 to +0.23, indicating spatial
variability but comparatively weaker contributions to LST modulation across the study area.
**Conclusions**
The manuscript unfolds by identifying the geospatial variations in crop residue–based
fires and their associated impacts on aerosol loading and land surface temperature across
northwestern India. A brief methodology and key findings are summarized in Fig. S8. Based
on year-wise, pixel-level fire intensity, the geographical region with intensive fire activity was
initially delineated, and all satellite-derived and reanalysis datasets were subsequently
processed exclusively over the selected zone. A robust and consistent spatial correlation
between FRP, AOD, and LST was observed across multiple years, indicating potential fire-
induced perturbations in LST. The Hurst exponent analysis reaffirmed the long-term
persistence of fire intensity, surface temperature, and aerosol loading time series. A grid-
based analysis over the intensive fire zone revealed a significant increase in both LST and AOD
during the peak fire season.
The article further employs the Random Forest model and Geographically weighted
regression (GWR) to assess the potential influence of FRP and aerosol loading on LST, while
accounting meteorological covariates, physical environment, vegetation characteristic and
surface property as confounding factors within the selected zone. Two contrasting scenarios
were hypothesized to examine the FRP–LST–AOD nexus. Scenario 1 considered spatially
aggregated FRP from fire initiation to subsidence, whereas Scenario 2 focused on days
characterized by high-intensity fires exhibiting a strong positive correlation between FRP and
LST. In both the scenarios, the Random Forest regression successfully captured and mapped
FRP-induced modulation of LST, though with varying magnitudes. A distinct increase in FRP-
induced LST modulation was observed during high-intensity fire events. Both boundary layer
height and columnar aerosol loading also contributed partially, with aerosols' influence on
LST increasing during periods of intense release of fire energy. The Global Moran's I test
indicated significant spatial clustering of LST while GWR results further confirmed FRP and
AOD-modulated LST variations across northwestern India, highlighting strong spatial
heterogeneity in FRP-AOD-LST nexus.
This analysis reveals that the biophysical effects of crop residue–based fires across
northwestern India can substantially influence the regional radiative budget by altering LST.
The magnitude of LST modulation, however, depends on fire intensity and feedbacks from
regional meteorology. This study provides novel insights into residue-based fire induced
surface temperature dynamics in a region where recurrent fires have been historically linked
primarily with deteriorating air quality in Delhi and its surroundings. The observation-driven
analysis offers a comprehensive understanding of LST responses to residue burning and helps
reduce uncertainties in fire-induced modifications of the radiative budget. Nonetheless,
uncertainties remain due to unaccounted agricultural feedbacks, limited temporal coverage,
retrieval uncertainty in geospatial datasets, and the complexity in aerosol–meteorology
interactions. The multifaced influence of fire aerosols and energy on regional climate through
rapid atmospheric and land surface adjustments, remains complicated at the global level. Our
findings underscore the need for Earth system model–based simulations to better quantify
climate feedbacks from crop residue burning. Besides, assessing the underlying mechanisms
of fire-energy-induced changes in evapotranspiration, the radiative effects of aerosols, fire–
aerosol–meteorology feedbacks, and incorporating additional proxies could further reduce
the uncertainty in estimating radiative impacts from residue burning.

**Acknowledgments**
Authors acknowledges the fund received from Banaras Hindu University under Institute of
Eminence grant (6031). AP acknowledges fund received from Department of Science and
Technology under INSPIRE fellowship (IF220684). Authors also acknowledge open source
software like R (V4.4), Python (V3.7) and QGIS (V3.28) for extracting and plotting the dataset.
**Data Availability**
All the data used in this analysis are available freely. VIIRS and MODIS data can be accessed
via NASA Earthdata (https://earthdata.nasa.gov), and ERA5 reanalysis data is available from
ECMWF Copernicus (https://cds.climate.copernicus.eu/). SMAP Soil moisture data is available
at https://nsidc.org/data/spl1ctb_e. All dataset were last accessed on November 13, 2025.
We thank NASA for providing the VIIRS and MODIS data, and the Copernicus Climate Change
Service (C3S) for the ERA5 reanalysis data.
**Authors contributions**
AP: Data curation, formal analysis and interpretation; RS: Data curation, formal analysis; KA:
Data curation, formal analysis; NC: Data curation, formal analysis; TB: conceptualization,
methodology and interpretation, funding as well as writing and editing manuscript.
**Competing interests.** Authors declare that they have no conflict of interest.
**Supporting Information.** The supporting tables (8) and figures (8) are included in
supplementary file.

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
