# Peer review of "Spatial influence of agricultural residue burning and aerosols on land surface"

_EGUsphere, 2025_

## Author Comment (AC1)

Title: Spatial influence of agriculture residue burning and aerosols on land surface temperature MS No.: egusphere-2025-3163

**Response to Referee # 1**

Authors are grateful to the reviewer for constructive comments and suggestions. In authors' response, we have responded point-by-point to each comment (reviewer's comments are in *blue* and authors' responses are in *black*), and have included the revisions in the text with track-change.

This manuscript try to address the relationship between fire radiative power (FRP), aerosol optical depth (AOD), and land surface temperature (LST) in northwestern India using multi-source remote sensing data combined with machine learning (random forest) and spatial regression (GWR). The topic is timely and relevant, particularly in the context of agricultural residue burning and its climatic impacts. The integration of multiple data sets and methods is commendable.

However, the current version has several shortcomings: the grammars and sentences are so poor, the transparency of data and methodology is limited, the interpretation of results is sometimes superficial and overly focused on correlations, and the discussion of mechanisms and uncertainties is insufficient. The conclusions also need to highlight the novelty and practical implications more clearly. With revisions to strengthen the grammars, methodological rigor, deepen interpretation, and improve clarity of presentation, this paper could make a valuable contribution. I recommend a major revision before it could be accepted.

**Major Comments**

The manuscript suffers from awkward sentence structures, grammatical errors, and weak logical transitions, which significantly reduce its readability and overall fluency. The authors should revise these basic problems as these have actually lowered the paper's quality. For example, it's hard to understand clearly without more following contexts when reading the first sentence in the abstract.

Thank you for all the valuable comments and suggestions. The revised manuscript has been thoroughly reviewed and refined to improve its clarity, language, grammar, and overall presentation. All suggested changes have been incorporated, including the addition of a new table (Table S1) summarizing the datasets and a workflow figure (Figure S1) in the supplementary material. Authors have updated the methodology of space-for-time, Random Forest and GWR by integrating additional datasets including meteorological covariates (PBLH, AT, SR, RH and PR), physical environment (elevation), vegetation and soil characteristics (NDVI, soil moisture), climatological mean LST and AOD, and surface property (albedo). The interpretation of the results has been strengthened, and the conclusion section has been comprehensively redrafted to enhance coherence and scientific rigor.

Clarity of research gap and contribution: The introduction should more clearly state why the FRP-LST relationship is poorly understood and what gap this study fills.

Authors have highlighted in the introduction that several studies have investigated the association between FRP and LST across diverse forest regimes but very few over crop land (included in L91–L102). Biomass burning broadly occurs in two forms: (i) large, high-intensity forest fires, and (ii) small, sporadic and spatially fragmented agricultural residue fires. Forest fires typically release substantial radiative energy, induce strong surface albedo and evapotranspiration changes, and therefore exert a pronounced influence on the regional radiation budget. Consequently, the fire–LST association has been well documented in forest-dominated ecosystems (e.g., Zhao et al., 2024; Alkama and Cescatti, 2016; Liu et al., 2018, 2019) and authors have discussed these findings in the introduction.

In contrast, residue-burning in croplands is sporadic, short-lived, and generally lower in fire intensity, which often results in weaker radiative forcing and a muted thermal response. As noted in L99–L107, such fires may not always generate a detectable modification in surface albedo or evapotranspiration, making their influence on LST more subtle and more difficult to isolate. Detecting this signal requires regions where residue burning occurs at sufficiently large spatial scales, such as northwestern India or parts of China. This additionally constrained by the limited availability of high-resolution datasets, such as AOD, LST, and FRP, as agricultural residue burning typically occurs over small and fragmented spatial scales.

The novelty of the study is included in L110-L127. Briefly, it states:

'....we addressed two key questions: (1) Does LST respond to changes in fire intensity over northwestern India? and (2) How do local meteorology and aerosol loading modulate LST variation with respect to space and time? To the best of our knowledge, this is the first systematic assessment of agricultural residue fire—driven modulations in LST over northwestern India. By integrating multiple geospatial observations, the analysis offers critical insights into the biophysical feedbacks of residue-based fire and advances understanding of LST responses to residue burning. Further, it refines estimates of fire-induced perturbations in the regional radiative budget offering valuable representation of biomass-based fire in Earth system models.'

Data description and transparency: Core datasets, resolutions, and time spans should be reported in the main text. Random forest and GWR parameter choices must be described for reproducibility. One table including all the datasets, period and their parameters, and a workflow on how to deal with the datasets and the following methods would be better to understand for both the reviewers and readers. Here is a reference: Figure 4 from https://doi.org/10.1016/j.rse.2025.114917.

All satellite-derived datasets used for the spatial analysis, including retrieval algorithms, spatial resolution, product version, uncertainty estimates (where reported in the literature), and quality flags are described in Section 2.2 ("Spatial Dataset"). In addition, following the reviewer's suggestion, we have included a new

supplementary table that provides a consolidated description of all core datasets, including data source, algorithm, temporal coverage, and spatial resolution (Table S1).

Timeframe considered for analysis is mentioned in section 2.1 (L143-145): It states: 'In this study, geospatial analyses of LST, fire activity, and aerosol loading were conducted over northwestern India during October—November between 2017 and 2021.' This is reemphasized in L238-240.

A table for RF parametrization is now also included as Table S6. Results of Global Moran's I Summary for LST across intensive fire zone for year 2017-2021 is included as Table S7. An improved table including GWR simulation criteria and model performance evaluation is included in supplementary file as Table S8.

A schematic workflow indicating datasets and methods is now included in supplementary as Figure S1.

Fig. S1. Schematic workflow indicating core datasets and adopted methodology.

Methodology: The two-scenario design is interesting but may introduce bias. Justification and limitations should be explained. Too many scales are used, so it's hard to compare between each other in a uniform The uncertainties should be discussed.

**# Justifications to two-scenario design:**

The onset, peak, and duration of post-harvest residue burning vary substantially from year to year, and numerous small, intermittent fires occur throughout the harvesting period. Unlike forest fires, residue-burning events are highly sporadic and exhibit strong spatial and temporal heterogeneity (Fig. S2 and S4) due to fragmented landholdings where individual fields may experience multiple low to high-intensity fires. Such small fires release insufficient radiative energy to meaningfully alter evapotranspiration or surface albedo and therefore have limited impact on LST. However, during peak burning periods, fire intensity increases markedly and has the potential to modify the regional radiative balance.

To examine the temporal dynamics of fire intensity and its implications for regional LST, two scenarios were defined using distinct thresholds. Both Scenario 1 and Scenario 2 were used to capture days with substantially elevated FRP across the region (Table S4). In Scenario 1, a relatively larger number of fire days were selected within each burning season, beginning from the initial rise in FRP and continuing until a marked decline in aggregate FRP was observed. All small, sporadic, and low-intensity fires occurring during the early and late stages of the burning season were deliberately excluded. Intermittent cases in which aggregate FRP increased by ≥50% relative to the preceding day but subsequently declined were also omitted. To avoid the inclusion of small-scale fire clusters, an additional criterion of cumulative FRP >1500 MW was applied. Scenario 2 included only periods characterized by persistently high FRP values that exhibited a strong positive association with regional mean LST. It represented days with a steady increase in aggregate FRP over time, indicating intensifying fire activity and energy release, accompanied by a positive correlation with regional mean LST. The requirement of >95% data completeness across FRP, LST, AOD, and meteorological variables restricted the number of eligible days per year but improved the robustness of the results.

Authors acknowledge that the major conclusions of this study are not sensitive to the exact threshold choices or to sensitivity checks. However, variations in thresholds can shift the yearly temporal window, leading to differences in the absolute magnitude of LST change across northwestern India. A comparison of the average FRP per day between Scenario 1 (12,152 MW/day) and Scenario 2 (18,054 MW/day) for representative years such as 2017 indicates a distinct difference in the total energy released from residue-burning events. Both scenarios consistently identify FRP as a key driver of LST variability; however, the exact magnitude of LST change is sensitive to the spatial region and the selection of fire-affected days. Regions experiencing high-intensity fires exhibit substantially greater increases in LST compared to areas with low-intensity or sparse fire occurrences. Furthermore, the extent of LST variation also modulated by prevailing meteorological conditions (like PBLH), as illustrated in Figure 8. Overall, our analysis provides compelling evidence that residue-based burning across northwestern India significantly influences LST and the regional radiative budget. Nevertheless, as emphasized in the abstract (L27-L29), the precise magnitude of LST perturbation associated with residue burning remains dependent on both fire intensity and concurrent meteorological conditions.

**# Selection of scales:**

All the satellite retrievals and reanalysis datasets were analyzed in two contexts: (1) grid-based analysis and (2) analysis using spatial means. Grid-based analysis was made to establish spatial correlation (Fig. 4d-f), to compute long-term persistence of data series using Hurst Exponent (Fig. 5), to quantify impact of FRP on LST and AOD (Fig. 7) following a space-for-time approach and GWR analysis (Fig. 9). The selected grid size and rationale of using such grids is mentioned in the text in section 2.4, 2.5 and 2.6.

Rest of the analysis like time-series evaluation (Fig. 6) and Random Forest (Fig. 8) were performed using daily-based spatially averaged means computed over intensive fire zone.

The uncertainties associated with the analysis have been addressed in the manuscript. It states:

L742–L746: 'Nonetheless, uncertainties remain due to unaccounted agricultural feedbacks, limited temporal coverage, retrieval uncertainty in geospatial datasets, and the complexity in aerosol–meteorology interactions. The multifaced influence of fire aerosols and energy on regional climate through rapid atmospheric and land surface adjustments, remains complicated at the global level.'

L606-L615: 'To quantify uncertainty in the estimated differences between fire-affected and non-fire-affected grid cells, we further computed 95% confidence intervals for  $\Delta$ LST and  $\Delta$ AOD using nonparametric bootstrapping. ...... Nonparametric bootstrapping results into significant increase in both  $\Delta$ LST (0.57°C; 95% CI: 0.33–0.81°C) and  $\Delta$ AOD (0.13; 95% CI: 0.08–0.17) in fire-affected regions. Because both CIs do not overlap zero, these differences are statistically robust and unlikely to be due to sampling variability'.

How can the authors explain well the relationships between fire and climate without using the climate model output and discussions on the aerosols?

The authors acknowledge that, in the absence of a climate model, the direct impacts of fire on climate variables cannot be quantitatively assessed. Accordingly, the revised manuscript avoids making any causal claims regarding direct fire—climate linkage. In principle, fire can influence climate not only through the emission of aerosols and trace gases but also by altering the exchange of carbon, water, and energy between the land surface and the atmosphere. Fire-induced changes in key biophysical properties, such as surface albedo, evapotranspiration, and sensible heat flux, modify the surface energy balance, with the absorbed energy dissipated through both radiative and non-radiative pathways. These considerations have now been incorporated into the Introduction to more accurately contextualize the scope and limitations of the study.

Results: Figures need clearer captions and higher resolution. Results should include approximate magnitudes (e.g., RMSE in °C).

Thank you for this observation. All figure captions have been rechecked for clarity, and the necessary corrections have been made. Regarding image resolution, the reduction likely occurred during the Wordto-PDF conversion; this will be addressed during the final submission stage. High-resolution images have

already been provided in the accompanying .doc file. Authors also apologize for the earlier omission, the magnitudes of all parameters, including RMSE and MAE (both in °C), are now included in the revised manuscript and specified in the corresponding figure captions.

Discussion: Too correlation-focused; mechanisms (direct heating, aerosol–radiation effects, meteorology) should be elaborated. AOD's nonlinear effect at high values needs more explanation.

Thank you for this comment. Accordingly, authors have improved the discussion. The essence of this study is to investigate the FRP–AOD–LST nexus over crop residue burning regions using a combination of satellite-based observations and reanalysis datasets. The temporal associations among FRP, AOD and LST clearly reveal the immediate response of aerosol loading and surface temperature to fire activity. To further quantify these dynamics, we employed a space-for-time approach to estimate changes in LST and AOD as a function of FRP. To ensure that changes in LST and AOD were attributable solely to fire activity, grids with similar characteristics in terms of topography (elevation), meteorological covariates (PBLH, AT, SR, RH and PR), vegetation and soil characteristics (NDVI, soil moisture), climatological mean LST and AOD, and surface properties (albedo) were compared with control grids. Authors further estimate 95% confidence intervals for  $\Delta$ LST and  $\Delta$ AOD due to residue-based fire. Nonparametric bootstrapping results into significant increase in both  $\Delta$ LST (0.57°C; 95% CI: 0.33–0.81°C) and  $\Delta$ AOD (0.13; 95% CI: 0.08–0.17) in fire-affected regions.

Subsequently, these relationships were modelled using a geospatial tree-based regression framework that integrates both temporal features (e.g., day-specific retrievals) and spatial predictors (e.g., regional meteorology, aerosol loading, and fire intensity) to characterize the FRP—AOD—LST interactions within the intensive fire zone. Finally, we identified the spatial variability in LST using GWR. The discussion combines insights from spatial associations among FRP, AOD, and LST with evidence derived from spatial modelling, time-series analysis, space-for-time analysis, and GWR, thereby providing a comprehensive understanding of fire-induced land—atmosphere interactions.

Initial submission

**Revised PDP plots**

The PDP plots submitted during initial submission has been updated in revision as authors integrates additional parameters including meteorological covariates (PBLH, AT, SR, RH and PR), physical

environment (elevation), vegetation and soil characteristics (NDVI, soil moisture), and surface property (albedo) in the Random Forest model.

Authors have now included explanations on aerosols' non-linear effect on LST in L670-L686. Text reads as: 'The estimated effects of both FRP and AOD exhibit a non-linear, saturating response. LST increases sharply at low-to-moderate values of each predictor but the effect progressively weakens at higher magnitudes, approaching an asymptotic limit. This behaviour likely arises from the complex interplay of radiative and thermodynamic processes associated with biomass-burning emissions. Fire-originated aerosols exert both direct and indirect radiative effects whose magnitudes and signs vary with aerosol loading and composition (Freychet et al., 2019; Xu et al., 2021; Tian et al., 2022). At moderate aerosol loading, UB-absorbing black carbon aerosols may enhance atmospheric heating and can transiently increase near-surface temperature (Jacobson, 2001). Fire-induced convective plumes may initially enhance surface temperatures, whereas strong aerosol build-up can reduce solar transmittance to the ground. Aerosol-cloud interactions further contribute to non-linearity by modifying cloud microphysics, lifetime, and albedo, altering the regional radiative balance. Additionally, aerosol-driven changes in boundary-layer structure, evapotranspiration, and soil moisture introduce additional land—atmosphere feedbacks. Together, these interacting processes operate across multiple spatial and temporal scales and do not scale linearly with aerosol loading or fire intensity, producing the observed non-linear LST response.'

Uncertainty and validation: Retrieval errors, short time series, and possible multicollinearity should be acknowledged. Additional trend tests could be considered.

Retrieval errors associated with each dataset are described in Section 2.2. Details on quality assurance, significance levels, and spatial resolution are provided in Section 2.2 and now, summarized in Table S1.

Authors have now also quantified uncertainty associated with yearly-increase in  $\Delta$ LST and  $\Delta$ AOD by fire by computing 95% confidence intervals. Nonparametric bootstrapping results into significant increase in both  $\Delta$ LST (0.57°C; 95% CI: 0.33–0.81°C) and  $\Delta$ AOD (0.13; 95% CI: 0.08–0.17) in fire-affected regions. As both confidence intervals lie entirely above (or below) zero, the differences can be considered statistically robust, with minimal likelihood that they arose from sampling variability.

The uncertainties associated with the analysis have been addressed in the manuscript. It states:

L742–L751: 'Nonetheless, uncertainties remain due to unaccounted agricultural feedbacks, limited temporal coverage, retrieval uncertainty in geospatial datasets, and the complexity in aerosol-meteorology interactions. The multifaced influence of fire aerosols and energy on regional climate through rapid atmospheric and land surface adjustments, remains complicated at the global level. Our findings underscore the need for Earth system model—based simulations to better quantify climate feedbacks from crop residue burning. Besides, assessing the underlying mechanisms of fire-energy-induced changes in evapotranspiration, the radiative effects of aerosols, fire-aerosol-meteorology feedbacks, and incorporating additional proxies could further reduce the uncertainty in estimating radiative impacts from residue burning.'

Section 2.7: Multicollinearity was evaluated using the Variance Inflation Factor (VIF). The assessment using VIF explained an acceptable level of multicollinearity (VIF<5) which allows authors to perform Random Forest. All biophysical, land-surface, and meteorological variables met acceptable VIF thresholds, except solar radiation, which was therefore excluded from Random Forest and GWR analysis. Trend analysis was not performed due to limited temporal extent of the dataset. It is mentioned in the text.

Conclusions: Should better emphasize novelty and implications for residue burning management and regional climate. Future directions could be more concrete.

The conclusion section has been redrafted with greater emphasis on the novelty, key findings, associated uncertainties, and possible implications of the findings. Discussion on residue burning management has been kept brief, as it lies beyond the scope of this study and has been extensively covered in previous publications addressing effective agricultural management practices to reduce farmers' reliance on residue burning.

Additionally, the section now includes updated future research directions. The text reads as:

'The multifaced influence of fire aerosols and energy on regional climate through rapid atmospheric and land surface adjustments, remains complicated at the global level. Our findings underscore the need for Earth system model—based simulations to better quantify climate feedbacks from crop residue burning. Besides, assessing the underlying mechanisms of fire-energy-induced changes in evapotranspiration, the radiative effects of aerosols, fire—aerosol—meteorology feedbacks, and incorporating additional proxies such as boundary layer height and soil moisture could further reduce the uncertainty in estimating radiative impacts from residue burning.'

**Minor Comments**

The titles of both manuscript and supplementary are different, besides "agriculture residue burning" should be "agricultural residue burning" or "crop residue burning", please keep the same and double check before uploading.

Apology for this error. Modified the title in revised supplementary file. It now reads as: *Spatial influence of agricultural residue burning and aerosols on land surface temperature*.

Use more formal academic expressions instead of colloquial wording. Language editing for conciseness and precision.

Thank you for this comment. Accordingly, extensive language editing has been made throughout the manuscript to improve readability, grammar, and conciseness.

L10, 28, "The biophysical effect of agriculture-residue based fire through excessive release of energy and carbonaceous aerosols essentially unaccounted globally" and the last sentence (L28) as and the sentences should be organized to state your meanings clearly.

Modified accordingly.

Now the abstract starts with the sentence: 'The biophysical effects of agricultural residue burning, driven by the excessive release of energy and carbonaceous aerosols, remain poorly quantified at the global scale.'

Abstract ends with: 'It further highlights that the magnitude of this perturbation is governed by interannual variations in fire intensity and influenced strongly by prevailing meteorological conditions.'

L23, can the authors explain more on the how 'significant', which is vague for the readers. Avoid using 'significant' without numbers added.

The word 'significant' is now excluded from the revised text as variations in relative feature importance were not assessed statistically.

However, in revised submission, authors have now quantified uncertainty in the estimated differences between fire-affected and non-fire-affected grid cells by computing 95% confidence intervals for  $\Delta$ LST and  $\Delta$ AOD. Nonparametric bootstrapping results into significant increase in both  $\Delta$ LST (0.57°C; 95% CI: 0.33–0.81°C) and  $\Delta$ AOD (0.13; 95% CI: 0.08–0.17) in fire-affected regions. Because both CIs do not overlap zero, authors made the following statement in abstract:

L18-L20: 'Over intensive fire zone, a space-for-time approach revealed significant increase in both  $\Delta$ LST (0.57°C; 95% CI:0.33-0.81°C) and  $\Delta$ AOD (0.13; 95% CI:0.08–0.17) due to fire'.

L25, Geographically Weighted Regression should be "Geographically weighted regression (GWR)". Complied accordingly.

L56, add the official reference of "4.1 million ha".

**Reference included as:**

NAAS, National Academy of Agricultural Sciences, 2017. Innovative Viable Solution to Rice Residue Burning in Rice-Wheat Cropping System through Concurrent Use of Super Straw Management System-fitted Combines and Turbo Happy Seeder", New Delhi.

L65-66, "with roughly 20-25% i.e. 100-120 MT/yr residues usually burn in the field itself, majority (~20-25 MT/yr) of such practised over northwest Gangetic plain", this is not a good sentence for people to understand what the authors want to share, please find some porfessional people to help refine it.

We apologize for the lack of clarity in the original text. The statement has been redrafted for clarity and now reads as:

'India produces an estimated 500 million metric tonnes (MT) of crop residues annually, of which 20–25% are disposed of through open-field burning. Crop residue burning is particularly prevalent in northwestern

India, where roughly 20-25 MT of residues are set on fire each year (Balwinder-Singh et al., 2019; Lan et al., 2022)'

**L70, do the authors really think that the fire occurrences will increase the vegetation index? Present why.**

We apologize for the lack of clarity in the original text.

The authors clarify that crop production across northwestern India has increased steadily over time, as indicated by consistent rises in vegetation indices. Concurrently, the region has experienced an increase in fire frequency and atmospheric aerosol loading, suggesting a strong temporal linkage between agricultural intensification and biomass burning.

**Revised text reads as:**

'Notably, the frequency of fire incidences has exhibited a persistent upward trend, coinciding with concurrent increases in vegetation indices and atmospheric aerosol loading (Vadrevu et al., 2019; Jethva et al., 2019).'

**L71, Beside -> Besides**

Modified. The sentence has been updated and mentioned in the following comment.

**L73-74, 'be it' is this the right grammar? And why there is a commar after the prepositions, such as thereby and however in the middle of the sentences.**

We apologize for the oversight in the original text. The correction has now been made in the revised manuscript. The text now reads as:

'In addition to atmospheric emissions, fires exert numerous biophysical impacts on the surrounding ecosystems. Fire induces a cascade of consequential processes, including modifications to the surface energy balance, redistribution of nutrients, alterations in species composition, changes in surface albedo, and variations in evapotranspiration rate (Ward et al., 2012; Liu et al., 2019).'

**L82, add references to support the authors' statements.**

Appropriate references (Lasko and Vadrevu, 2018; Jethva et al., 2019; Chuvieco et al., 2021; Aditi et al., 2025) have been added to support the statement (L84-85).

**L86, is surface albedo a process, or its changing?**

Surface albedo is a biophysical variable and now corrected in text. Over an agricultural land, surface albedo can vary depending on vegetation growth stage, and due to variations in soil moisture, precipitation and temperature. Surface albedo also changes after a fire because fire removes, darkens, or replaces the reflective components of the land surface. Fire also produces charcoal, black carbon, and ash that settle on soil surface and changes Its reflectance.

**L90, enhance is a verb, enhancement?**

Thank you for pointing out this error; it has now been corrected in the revised version.

**L93, evident should be a verb?**

Thank you for pointing out this error; it has now been corrected in the revised version.

**L97, agriculture farmland-> agricultural farmland, agriculture residue burning -> agricultural residue burning**

Corrected in each instance in revised text.

L101, cop?

It should be crop, now corrected.

**L102, what remained valid till 1~3 day?**

Zhang et al. (2020) has reported that the influence of crop residue burning on LST in three provinces across China, existed for 1-3 days and did not disappear immediately.

**L108, the reanalysis is model output, do the authors mean reanalysis is treated as observations here?**

Apology for this error. Only satellite-based datasets were considered as observation. The text has been corrected for clarity.

L111, Several statistical means were explored -> Several statistical methods were applied? Corrected.

L124, food grain generation-> food grain production

Modified.

**L137, please use the dotted line when ploting the disputed boundary in the small figure, especially between India and its neighbouring coutries such as China, Pakistan and etc.**

The authors greatly appreciate the reviewer's concern regarding the depiction of disputed international boundaries. However, as per national guidelines and instructions from authors' institute (Banaras Hindu University), authors are bound by laws to use maps officially approved by the Government of India for all research publications. For transparency, a Publisher's Note indicating neutrality with respect to jurisdictional claims in the text and figures will be included.

L176, the defination of LST (Land surface radiometric temperature) is conflict with the one (Land surface temperature) in L89.

Thank you for this note, accordingly the definition of LST has been amended in section 2.2.

L185, a uncertainty-> an uncertainty

Corrected.

L185-186, Both daytime maximum and nighttime minimum LST approximately at 1:30 PM and 1:30 AM local time respectively, are available. -> The dataset provides daytime maximum LST (at 1:30 PM local time) and nighttime minimum LST (at 1:30 AM local time).

Thank you for this note, accordingly the text has been updated.

**L187-189, any relationships between the previous sentence and this one, why use however?**

The authors would like to emphasize that among the available daytime and nighttime LST datasets, the daytime (1:30 PM local time) LST from Aqua MODIS was selected, as it coincides with the VIIRS overpass and represents the period when crop residue—based fires are expected to peak.

Revised text now reads as:

'Here, daytime LST dataset were obtained solely from the MODIS sensor onboard the Aqua satellite to closely coincide with VIIRS fire count observations at 1:30 PM local time, a period when crop residue—based fires are expected to reach at peak.'

**L194, can the authors explain more on $\pm (0.05+20\%)$ ?**

Sayer et al. (2019) reported an estimated error of  $\pm$ (0.05+20%) in VIIRS Version 1 DB AOD dataset when compared globally with AERONET AOD. This indicates the uncertainty associated with VIIRS AOD data. The uncertainty value  $\pm$ (0.05 + 20%) refers that the VIRS DB AOD may vary by a fixed absolute component ( $\pm$ 0.05) plus a relative (percentage) component ( $\pm$ 20% of the measured value) against AERONET AOD. This simply refers to VIIRS AOD could range from 0.75 to 1.25 for a concurrent AERONET AOD of 1.0, on a global basis. Its noteworthy that VIIRS DB AOD performs well over bright surface including agriculture land while performs relatively poor over dense vegetation.

**L197, why not use a ROI to define the "selected spatial domain"?**

One of the key novelties of this research lies in the use of a spatially varying agricultural field area determined based on year-specific fire pixel density. As presented in Table S2 and described in Section 2.3.1 (Fig. 2), a spatial domain was delineated to capture the potential effects of day-to-day variations in fire intensity and aerosol loading on LST. This domain was defined using a threshold fire radiative power density on yearly basis. Consequently, no fixed ROI was employed for data retrieval and geospatial analysis; instead, the analysis adjusted to the spatial extent of active fire occurrences for each year.

L284, fulfilling -> to fulfill.

Corrected.

Figure captions: combine it and the NOTE as one caption according to the journal's requirement.

Complied, figure caption and NOTE is now combined in each figure.

L525, format the caption of Fig. 7 as previous ones.

Corrected.

L 535-538, this sentence should be moved into Section 2.8.

Complied, the sentence is now placed in section 2.8.

L561-564, I am confused if there are any evidences or results to show the relationships between FRP, LST and the regional climate/human health. This could be discussion, but this can not be the result without any evindences shown.

Authors acknowledge that the present study does not investigate fire-induced changes in regional climate or human health. However, since the results and discussion were drafted together, the statement was included to ensure continuity in the narrative. It highlights that fire emissions can exert an immediate influence on the regional radiative budget (through changes in LST) and can also degrade air quality, thereby potentially impacting human health. The statement is now changed and reads as:

'Fire radiative power emerged as the dominant predictor under both scenarios, indicating the strong influence of fire-related energy release on regional radiative balance, likely through reduced evapotranspiration and fire-induced changes in surface albedo (Liu et al., 2018, 2019).'

L592, GWR has been defined.

Complied.

L607, the statements of 'fire impact the regional climate' are not strong based on the results as only some discussions are shown in the manuscript.

Complied, the statement now reads as:

'This analysis reveals that the biophysical effects of crop residue—based fires across northwestern India can substantially influence the regional radiative budget by altering LST.'

The conclusion is overly repetitive and needs to be reorganized and easy-understanding. Usually, in the Conclusion part, no more references are needed.

Thank you for this suggestion. Accordingly, conclusion is redrafted emphasizing novelty, major findings, uncertainty, and future directions. References are now removed from the conclusion.

Evidence of change in LST due to biomass burning, as reported in literature, is now moved to section 3.4.

Improve cross-references between text and figures. References are not well cited in the EGU style, please read https://www.atmospheric-chemistry-and-physics.net/submission.html carefully and revise them.

All the references were modified according to EGU style as per *Atmospheric Chemistry and Physics* journal. In-line citations were also cross-checked for accuracy.

\*\*\*

---

## Author Comment (AC2)

Title: Spatial influence of agriculture residue burning and aerosols on land surface temperature MS No.: egusphere-2025-3163

**Response to Referee # 2**

Authors are grateful to the reviewer for constructive comments and suggestions. In authors' response, we have responded point-by-point to each comment (reviewer's comments are in *blue* and authors' responses are in *black*), and have included the revisions in the text with track-change.

Pandey et al.'s study "Spatial influence of agriculture residue burning and aerosols on land surface temperature" presents an observation-driven study of how crop-residue fires in northwest India influence land surface temperature (LST) and aerosol loading. They identify year-specific intensive fire zones using VIIRS FRP, retrieve VIIRS AOD and MODIS Aqua daytime LST, and use AgERA5 meteorology to control for meteorological context. They apply a space-for-time grid comparison to estimate ΔLST and ΔAOD associated with fire, compute Hurst exponents for persistence, and develop two Random-Forest (RF) regression scenarios (broad fire season and high-correlation windows) to quantify relative feature importance. Finally, they run a Geographically Weighted Regression (GWR) of FRP and LST to map spatial heterogeneity. The paper reports an average fire-induced ΔLST ≈ +0.48°C (range −0.55 to 1.69°C) and ΔAOD ≈ +0.19 yr-1 during the peak season, and finds FRP is the top RF predictor of LST in both scenarios (with much higher RF performance in the "scenario 2" windows). Crop-residue burning in NW India and other parts of South Asia has major air-quality and climate implications, this study's focus on crop-burning and LST is important for this region. The use of VIIRS FRP, VIIRS AOD, MODIS LST, MODIS LC data and AgERA5 meteorology enables a multi-angle observational assessment. The space-for-time comparison, Hurst analysis, random forest for non-linear attribution, and GWR for spatial heterogeneity form a coherent methodological ensemble. However, there are some major concern and queries that needs to be properly addressed at this stage:

Thank you for the valuable comments and suggestions. All recommended changes have been incorporated, including improvements to the space-for-time methodology, Random Forest analysis, and GWR. Additional parameters including meteorological covariates (PBLH, AT, SR, RH, and PR), physical environment (elevation), vegetation and soil characteristics (NDVI, soil moisture), climatological mean LST and AOD, and surface properties (albedo), were included in the selection of "fire" and "no-fire" grids to strengthen the causal attribution of fire to  $\Delta$ AOD and  $\Delta$ LST. These parameters were also included in Random Forest and GWR. A nonparametric bootstrapping is performed to estimate uncertainty in  $\Delta$ AOD and  $\Delta$ LST. A new supplementary table (Table S1) summarizing all datasets and a workflow diagram (Figure S1) have been added. The interpretation of results has also been revised to improve coherence and scientific clarity.

1. LST is strongly influenced by near-surface air temperature, PBL height, soil moisture, recent precipitation, cloud cover, surface albedo and vegetation state (NDVI/LAI). Although AgERA5 meteorology

(At, Sr, Pr, RH) is included as one of the predictors, the manuscript does not convincingly demonstrate that the estimated ΔLST (and RF / GWR results) are not driven by meteorological covariates or systematic land-cover differences between "fire" and "no-fire" grids. Without stronger control for these confounders, the causal attribution "fire to AOD and LST" remains tentative. In the space-for-time comparison, conducting matched comparisons, for instance for each fire grid choose one or more no-fire grids matched by NDVI, elevation, distance to major urban areas, and climatological mean LST. This reduces bias from non-random spatial placement of fires. Propensity-score matching or simple stratified matching would help. Additional proxies including but not limited to PBL height, surface soil moisture, and in-situ atmospheric radiative impacts induced by the fire-emitted aerosols themselves used in the predictor set may help clarify this relationship and strengthen the findings. However, I welcome the authors to instead post a rationale on why not including these variables and this suggested approach may still suffice in relationship quantification.

Thank you for this suggestion. Accordingly, authors have considered additional parameters, including meteorological covariates (PBLH, AT, SR, RH and PR), physical environment (elevation), vegetation and soil characteristics (NDVI, soil moisture), climatological mean LST and AOD, and surface property (albedo), into the selection of "fire" and "no-fire" grids to strengthen the causal attribution of fire in  $\Delta$ AOD and  $\Delta$ LST. As suggested, we also applied a stratified matching technique using combinations of major confounders and conducted comparisons within strata to estimate the difference in LST and AOD between fire and no-fire grids. This refinement substantially strengthened the robustness of our estimates, revealing a consistent and statistically meaningful increase in both LST and AOD in every year due to recurrent fire.

ts Modified results with additional confounders

A detailed and modified space-for-time approach is now included in section 2.6. Briefly, it states:

'....To ensure that changes in LST and AOD were attributable solely to fire activity, grids with similar characteristics in terms of topography, climate, and physical environment were compared (Liu et al., 2019). To achieve this, daily datasets including meteorological covariates (PBLH, AT, SR, RH and PR), physical environment (elevation), vegetation and soil characteristics (NDVI, soil moisture), climatological mean LST and AOD, and surface property (albedo) were extracted over both fire and no-fire grids at a spatial resolution of  $10 \times 10 \, \mathrm{km^2}$ . ....... Fire and no-fire grids with comparable spatial characteristics were grouped into a single stratum, and a stratified matching technique was applied to generate multiple strata based on combinations of the selected confounders. Grids were retained only when differences in their physical environment, vegetation and soil characteristics, climate and land cover between fire and no-fire

conditions were smaller than the defined thresholds ( $\Delta$ elevation < 50 m;  $\Delta$ NDVI <0.05;  $\Delta$ soil moisture <0.05;  $\Delta$ albedo <0.05;  $\Delta$ LST <10.0;  $\Delta$ AOD <0.80). Comparisons were then made within strata containing grids of similar attributes to ensure that the observed variations in LST and AOD could be attributed solely to fire activity'.

Authors would also like to emphasize that the entire residue-burning zone in northwestern India follows similar agronomic practices, with comparable land characteristics, vegetation dynamics, and climatic conditions, as it lies within a single composite climatic zone. Consequently, only subtle variations in meteorological covariates (At, SR, RH, PT) and PBLH (SD:  $\pm 10~m$  to  $\pm 33~m$ , yearly) were observed across the grids. All selected grids were representative of croplands within the extended geographical region; therefore, distance from urban centers was not incorporated as an additional constraint. This choice is justified by the fact that agricultural emissions overwhelmingly dominate over anthropogenic urban sources in the post-monsoon season when major residue burning occurs. Columnar aerosol loading was included in the analysis; however, fire-emitted aerosols were not considered separately, as segregating fire-derived aerosols from background loading could introduce additional uncertainty. Authors have included a rationale on variable selection criteria in section 2.6:

'It is noteworthy that the grids were not classified based on meteorological covariates, as only insignificant variations were noted among the grids. The entire northwestern cropland experiences a relatively uniform background climate during October–November, including comparable boundary layer heights, with PBLH standard deviations ranging from ±10 m to ±33 m within a single fire season. The climatological mean LST and AOD were computed only for the pre-fire season, during which none of the grids experienced residue-burning activity. Furthermore, grids were not differentiated by slope or aspect, given the minimal topographic variation across the Gangetic Plain.'

2. Provide details on RF hyperparameter tuning (max\_depth, max\_features, min\_samples\_leaf). The manuscript uses n\_estimators=100 with a fixed seed — please show whether you tuned parameters (grid search / CV) or at least show sensitivity to n\_trees and max\_features. To further imrpve RF model valiadtion, spatial and temporal block cross-validation (e.g., leave-one-year-out, or K-fold blocking by contiguous spatial clusters) and report cross-validated R2, RMSE, MAE. This may provide more robust predictive skill.

Thank you for this note and guidance. This has indeed improved model performance and creates a statistically rigorous and computationally efficient modelling outcome.

In the revised manuscript, authors have incorporated additional predictors, including FRP, AOD, regional meteorology, surface properties, and vegetation characteristics, into the Random Forest (RF) model to establish a non-linear statistical association between LST and multiple predictors. Accordingly, Section 2.8 (in methods) and Section 3.5 (in results and discussions) have been updated and expanded.

Key RF hyperparameters (n\_estimators, max\_depth, min\_samples\_split, min\_samples\_leaf, and max\_features) were optimized using Bayesian optimization implemented via BayesSearchCV in scikit-optimize. Bayesian optimization provides an adaptive and probabilistic search strategy that efficiently explores high-dimensional hyperparameter spaces, outperforming traditional grid and random search approaches in both accuracy and computational efficiency (Snoek et al., 2012; Shahriari et al., 2016; Frazier, 2018).

To ensure robust model evaluation and minimize temporal dependence, authors adopted temporal block cross-validation using a 3-fold GroupKFold strategy in scikit-learn, in which all samples from the same year were assigned to the same fold, following the blocking principles recommended by Roberts et al. (2017) and Valavi et al. (2019) for temporally structured datasets. This approach provides temporally independent estimates of predictive skill and mitigates information leakage across folds.

Following reviewer's recommendations, cross-validated R2, RMSE, and MAE is now reported in manuscript (Fig. 8), averaged across folds to provide an unbiased estimate of predictive accuracy. This combined framework, Bayesian hyperparameter optimization and temporally independent cross-validation, offers a statistically rigorous and computationally efficient modelling strategy. Details of the RF hyperparameter tuning procedure are included in Table S6.

Initial results

Modified results with RF hyperparameter tuning

3. The GWR model for scenario 2 is using only FRP, SR and AOD as predictor for LST, I do not understand the rationale of leaving out other local factors, included but limited to those mentioned in point 1 above. Are the authors testing the concept of using these specific variables exclusively in relationship to LST? However, I am confused if other meteorological variables and aerosol types (their optical variablility in terms of scattering and absorption, and how these may influence atmospheric heating/radiative forcing and near-surface based cooling/radiative forcing (Freychet et al 2019; Tiwari et al. 2023) and surface albedo (Hou et al. 2025) when running GWR could bias the local coeffcients. Local coefficients maybe absorb the effect of omitted spatially-varying covariates. I am confused why scenario 2 is missing out important variables. Adiitionally, please also include bandwidth and kernel details of the AICc minimization you mention.

Thank you for this suggestion. In the revised manuscript, all predictor variables used in the Random Forest model (AOD, PBLH, AT, RH, SR, PT, NDVI, elevation, albedo, and FRP) were also incorporated into the GWR framework. However, aerosol types were not included. Aerosols over South Asia exhibit substantial compositional diversity and are influenced by multiple mixed sources, and classifying aerosol type using AOD–fine-mode fraction–SSA combinations can introduce considerable uncertainty. Moreover, based on

our earlier trials using satellite datasets, carbonaceous smoke aerosols were the only type that could be identified with reasonable confidence over the northwestern region during the biomass-burning period. Including aerosol type as a predictor would therefore risk adding noise and misleading spatial patterns.

Accordingly, the local coefficient maps have been updated, the revised FRP–LST and AOD-LST GWR outputs for both scenarios are now presented in Fig. 9, while coefficient maps for the remaining predictors are provided in Fig. S7. The GWR model exhibited strong explanatory performance, with global R² values exceeding 0.74, indicating that the selected predictors effectively captured spatial variability in LST. The optimal bandwidth was determined via an iterative optimization procedure that minimizes the corrected Akaike Information Criterion (AICc). A new table (Table S8) has been added to the Supplementary Material, summarizing the GWR simulation setup, performance metrics, kernel structure, and bandwidth parameters used in the AICc minimization.

4. I am also confused with the descritption of scenario 2, specifically if the reported relative feature importance (RFI) is normalized in the right way? As you mention this is a normalized metric. But for scenario 2 FRP was 0.503 SR was 0.143 and Aerosol loading was 0.68. For these three predictors the normalized RFI sum more than 1. Is this a typographical error, a misunderstanding on my part, or is there some calculation mistake?

The authors apologize for this typographical error. The reported relative feature importance (RFI) values were, in fact, normalized.

In the revised manuscript, the RFI scores have been updated following the inclusion of additional parameters, and all reported values have been thoroughly rechecked to ensure accuracy.

5.  $\Delta$ LST is reported as +0.48°C (mean) with range, but it's unclear whether this difference is statistically significant after accounting for temporal variability and dependence, and how many grid cells underpin the estimates. Provide confidence intervals (e.g., bootstrapped CIs) for  $\Delta$ LST and  $\Delta$ AOD. Additionally, consider how comparison of pre-post events within the same grid for fire vs. similar non-fire grids) could help strengthen causual inference.

Thank you for this suggestion. In the revised manuscript, we applied nonparametric bootstrapping to assess whether the  $\Delta$ LST and  $\Delta$ AOD attributable to fire remained statistically significant after accounting for temporal variability and dependence (L606-L615). The grid selection criteria were further refined to ensure that the estimated changes in LST and AOD could be attributed solely to fire. A total of 7,489 paired fire and no-fire grid cells from 2017–2021 were used to quantify relative differences. All grids, regardless of fire status, were selected from within the extended geographical region to capture localized variations in surface temperature and aerosol loading.

To quantify uncertainty, we computed 95% confidence intervals (CIs) for  $\Delta$ LST and  $\Delta$ AOD using nonparametric bootstrapping. For each variable, 10,000 bootstrap samples were generated by resampling

grid pairs with replacement, and the mean difference was recalculated for each iteration. The  $2.5^{th}$  and  $97.5^{th}$  percentiles of the resulting sampling distribution were used to define the 95% CI. Bootstrapping revealed a statistically significant increase in both  $\Delta$ LST (0.57 °C; 95% CI: 0.33–0.81 °C) and  $\Delta$ AOD (0.13; 95% CI: 0.08–0.17) in fire-affected regions. As both confidence intervals exclude zero, the estimated differences are statistically robust and unlikely to arise from sampling variability.

6. Justify selection of FRP density threshold (>5 MW grid $^{-1}$ ), the 1500 MW threshold and the 50% growth/decline rule for scenario 1, and the r >= 5 threshold for scenario 2. Add rationale and sensitivity checks (e.g., try thresholds (+20%, -20%).

The onset, peak, and duration of post-harvest residue burning vary substantially from year to year, and numerous small, intermittent fires occur throughout the harvesting period. Unlike forest fires, residue-burning events are highly sporadic and exhibit strong spatial and temporal heterogeneity (Fig. S2 and Fig. S4) due to fragmented landholdings, where individual fields may experience multiple low-intensity fires. Such small fires release insufficient radiative energy to meaningfully alter evapotranspiration or surface albedo and therefore have limited impact on LST. However, during peak burning periods, fire intensity increases markedly and has the potential to modify the regional radiative balance.

To examine the temporal dynamics of fire intensity and its implications for regional LST, two scenarios were defined using distinct thresholds. Both Scenario 1 and Scenario 2 were used to capture days with substantially elevated FRP across the region (Table S5). In Scenario 1, a relatively larger number of fire days were selected within each burning season, beginning from the initial rise in FRP and continuing until a marked decline in aggregate FRP was observed. All small, sporadic, and low-intensity fires occurring during the early and late stages of the burning season were deliberately excluded. Intermittent cases in which aggregate FRP increased by ≥50% relative to the preceding day but subsequently declined were also omitted. To avoid the inclusion of small-scale fire clusters, an additional criterion of cumulative FRP >1500 MW was applied.

Scenario 2, by contrast, included only periods characterized by persistently high FRP values that exhibited a strong positive association with regional mean LST. It represented days with a steady increase in aggregate FRP over time, indicating intensifying fire activity and energy release, accompanied by a positive correlation with regional mean LST. The requirement of >95% data completeness across FRP, LST, AOD, and meteorological variables restricted the number of eligible days per year but improved the robustness of the results.

Authors acknowledge that the major conclusions of this study are not sensitive to the exact threshold choices or to sensitivity checks. However, variations in thresholds can shift the yearly temporal window, leading to differences in the absolute magnitude of LST change across northwestern India. Accordingly, as stated in the abstract, both scenarios consistently identify FRP as a dominant driver of LST variability, although the precise magnitude of LST perturbation remains sensitive to the domain and the selection of fire-affected days. Overall, this analysis provides robust evidence that residue burning across

northwestern India significantly influences LST and alters the regional radiative budget. Nevertheless, the exact magnitude of fire-induced LST perturbations depends on both the intensity of burning and concurrent meteorological conditions.

7. The Hurst exponent computed and interpreted as persistence (> 0.5), is relevant when there is large number of data points which are specifically not impacted by seasonal trends, however, in this case, with only 5-year dataset and strong seasonality, Rescaled Range (R/S) analysis for Hurst estimation can be sensitive to trend and seasonality. This is an important featured previously determined by various observational studies in this part of the world where both inter- and intra-annual variability is common (Lin et al. 2020; Liu et al. 2024 etc.). Did the authors conduct detrended fluctuation analysis (DFA) or remove seasonal cycle before computing Hurst. Furthermore, the author's interpret values of H > 0.5 as indicating persistence and suggest that anomalies may "remain stable in the near future." While H > 0.5 indeed indicates statistical persistence or long-term data analysis, this interpretation could overstate the predictive implications of the Hurst exponent, especially given the relatively short five-year data record and the presence of strong seasonal cycles (such as monsoon and agricultural seasonality) inherent in the dataset. I recommend the authors temper the predictive language by replacing claims that anomalies "will" persist with the more cautious and appropriate statement that H > 0.5 indicates statistical persistence. Additionally, the authors are encouraged to clarify whether seasonal cycles were accounted for or removed prior to computing the Hurst exponent, as this can significantly affect estimates derived from R/S analysis.

Thank you for this valuable comment. The authors did not apply detrended fluctuation analysis because the dataset used to estimate the Hurst exponent represents a single season (October–November) from 2017 to 2021. All residue-burning events examined in this study occur exclusively during the postmonsoon period, which exhibits distinct characteristics compared with the monsoon (JJAS) and winter (DJF) seasons. Therefore, the retrieval and analysis of FRP, AOD, and LST were based on a single-season dataset, and seasonal decomposition was not intended.

As suggested, the interpretation of the Hurst exponent for LST, FRP, and AOD in Section 3.2 has been revised. We no longer refer to "certainty" in predicting anomalies and instead emphasize statistical persistence within the dataset. This clarification has been incorporated into the revised manuscript.

8. There are several small typos/grammatical slips (e.g., "Dring" typo of "During" (Page 19), "reginal" typo of "regional" (Page 19 3.4), please go through the manuscript carefully and correct these and similar mistakes.

Thank you for this comment. Accordingly, extensive language editing has been made throughout the manuscript to improve readability, grammar, and conciseness.

\*\*\*\*\*\*

---

## Author Response (AR2)

Manuscript Number: egusphere-2025-3163                    Date: January 2, 2026
Title: Spatial influence of agriculture residue burning and aerosols on land surface temperature

Dear Prof. Jason Cohen
Editor
*Atmospheric Chemistry and Physics*

Authors sincerely acknowledge the comments provided by the Editor. The justifications in response to editor's comment is included here.

**EC: 1.** *The first is that a reviewer recommended some changes to how you present an overview of the workflow, and the second is related to the map in Figure 1. …..One reviewer has raised a concern regarding the depiction of international boundaries in the provided map. As an international journal, we maintain a policy of neutrality on territorial disputes…..*

*Please revise as follows:*

*1. Option A (Preferred): Replace the current map with one sourced directly from a standard, publicly available basemap (e.g., Natural Earth, OpenStreetMap, or UN Cartographic Section basemaps). Ensure that the exact source is cited.*

*2. Option B: If there is a specific, defensible scholarly reason for using the current boundary depiction (e.g., it is integral to the analysis of a specific dataset that uses those boundaries), the you must add a disclaimer note to the figure caption. This note should say something along the lines of: "The boundaries shown on this map are for illustrative purposes based on the source data [cite the source]. They do not imply endorsement of acceptance by the journal or publisher of any particular political or legal status of the territories depicted."*

**AC 1:** The authors greatly appreciate the Editor's concern regarding the depiction of disputed international boundaries. As per national guidelines and instructions from authors' institute (*Banaras Hindu University*), authors are bound to use maps officially approved by the Government of India for all research publications. Authors have used the India map from *Survey of India*' archive which is the official repository of maps in India.

For transparency, a Publisher's Note is included in Fig. 1: "The international boundaries shown on this map are for illustrative purposes based on the '*Survey of India*' archive. They do not imply endorsement of acceptance by the journal or publisher of any particular political or legal status of the territories depicted."

**EC: 2.** *Another reviewer suggestion which I support is to add a summary figure to help the community to summarize and follow what was done, what was quantified, and the pathways you used to accomplish this.*

**AC: 2.** Authors have now included a summary figure (Fig. S8, supplementary file) that illustrates the overall workflow of the study, highlighting the key methodological steps, quantified variables, and analytical pathways employed to achieve the study objectives. This figure provides a concise overview of the entire research framework and is intended to facilitate reader comprehension of the study design and outcomes.

Thank you very much for the opportunity to improve our submission.

Kind regards
*Tirthankar Banerjee*
Author